# In situ structure and dynamics of an alphacoronavirus spike protein by cryo-ET and cryo-EM

Cheng-Yu Huang [1,6], Piotr Draczkowski [1,2,6], Yong-Sheng Wang [1,3,6], Chia-Yu Chang [1,4,6], Yu-Chun Chien [1,3], Yun-Han Cheng [4], Yi-Min Wu[5], Chun-Hsiung Wang[5], Yuan-Chih Chang [1,5], Yen-Chen Chang [4], Tzu-Jing Yang [1,3], Yu-Xi Tsai [1,3], Kay-Hooi Khoo [1,3], Hui-Wen Chang [4] & Shang-Te Danny Hsu [1,3] ✉

Porcine epidemic diarrhea (PED) is a highly contagious swine disease caused by porcine epidemic diarrhea virus (PEDV). PED causes enteric disorders with an exceptionally high fatality in neonates, bringing substantial economic losses in the pork industry. The trimeric spike (S) glycoprotein of PEDV is responsible for virus-host recognition, membrane fusion, and is the main target for vaccine development and antigenic analysis. The atomic structures of the recombinant PEDV S proteins of two different strains have been reported, but they reveal distinct N-terminal domain 0 (D0) architectures that may correspond to different functional states. The existence of the D0 is a unique feature of alphacoronavirus. Here we combined cryo-electron tomography (cryo-ET) and cryo-electron microscopy (cryo-EM) to demonstrate in situ the asynchronous S protein D0 motions on intact viral particles of a highly virulent PEDV Pintung 52 strain. We further determined the cryo-EM structure of the recombinant S protein derived from a porcine cell line, which revealed additional domain motions likely associated with receptor binding. By integrating mass spectrometry and cryo-EM, we delineated the complex compositions and spatial distribution of the PEDV S protein N-glycans, and demonstrated the functional role of a key N-glycan in modulating the D0 conformation.

Coronaviruses (CoVs) are enveloped, positive-sensed single-stranded RNA (+ssRNA) viruses with a genome size that ranges between 26 and 32 kb. CoVs can be divided into four genera, namely the alpha-, beta-, gamma- and deltacoronavirus[1]. As a member of the alphacoronavirus genus, porcine epidemic diarrhea virus (PEDV) infects intestinal epithelial cells of pigs and causes digestive disorders such as watery diarrhea, vomiting, anorexia, and even death as a consequence of dehydration[2]. The first outbreak of PEDV was reported in the United Kingdom in the 1970s and subsequently spread to other European and Asian countries[2–4]. During the past decades, infections of the genotype I (G1) PEDVs have become an endemic swine disease in the Eurasia continent, with sporadic outbreaks characterized by relatively low

[1]Institute of Biological Chemistry, Academia Sinica, Taipei 11529, Taiwan. [2]Faculty of Pharmacy, Medical University of Lublin, ul. W. Chodzki 4a, 20-093 Lublin, Poland. [3]Institute of Biochemical Sciences, National Taiwan University, Taipei 11529, Taiwan. [4]Graduate Institute of Molecular and Comparative Pathobiology, School of Veterinary Medicine, National Taiwan University, Taipei 10617, Taiwan. [5]Academia Sinica Cryo-EM Center, Academia Sinica, Taipei 11529, Taiwan. [6]These authors contributed equally: Cheng-Yu Huang, Piotr Draczkowski, Yong-Sheng Wang, Chia-Yu Chang. ✉e-mail: sthsu@gate.sinica.edu.tw

morbidity and low mortality[2,5]. However, new variants of PEDV, which are classified into the genotype II (G2), have emerged in China since 2010 and rapidly spread across Asia and North America[6,7]. In addition to higher virulence and enteropathogenicity, porcine epidemic diarrhea (PED) symptoms caused by G2 PEDVs have high morbidity in all ages; it is particularly lethal in neonatal piglets from either naïve or vaccinated herds[7-9]. To date, G2 PEDVs have claimed millions of piglets worldwide, resulting in great economic losses[10,11]. Preventions and treatments of G2 PEDV infection remain very challenging, not least because of the limited knowledge of the mechanism of immune escape, immunogenicity, and antigenicity.

Like all coronaviruses, the highly glycosylated spike (S) protein of PEDV is responsible for virus entry, enteropathogenicity, and virus neutralization[2]. The PEDV S consists of an S1 subunit, an S2 subunit, a transmembrane domain (TM), and a cytosolic tail. The S1 subunit can be subdivided into three domains and two subdomains, namely the domain 0 (D0), the N-terminal domain (NTD), the subdomain 1 (SD1), the C-terminal domain (CTD), and the subdomain 2 (SD2)[12]. The D0 modulates the enteric tropism of PEDV by binding to sialic acids (SAs) on the surface of enterocytes[13-15]. The NTD is thought to mediate the interaction between the S protein and host cells by binding to the cell surface carbohydrate in most coronaviruses[16-18]; however, their specific functions in PEDV remain controversial[19,20]. The CTD is proposed to function as a receptor-binding domain (RBD), although the identity of the receptor remains to be established[12,16]. Despite earlier reports that suggest the porcine aminopeptidase N protein (pAPN, also known as CD13) to be a PEDV receptor[21-23], more recent findings observed PEDV infections in pAPN-knockout cell lines and pAPN-knockout pigs, suggesting that pAPN is not a bona fide PEDV receptor[20,24,25]. Nevertheless, continued efforts have led to the identification of several neutralizing epitopes in PEDV S of G1 and G2 strains[26-28], which may serve as the main targets for the vaccine developments and therapeutic strategy[29].

Recently, two distinct cryo-EM structures of the S proteins from two different PEDV strains have been reported. The S protein of the PEDV CV777, the popular parental vaccine strain of G1 PEDV, exhibits an "up" conformation for the D0[12], which is similar to the structure of the S protein of feline infectious peritonitis virus (FIPV)[30]. In contrast, the S protein of the PEDV USA/Colorado/2013 (CO/13) strain, a highly virulent G2 PEDV strain circulating in the USA, exhibits a "down" conformation for the D0[31], which is similar to the structure of the S protein of human alphacoronavirus NL63 (HCoV-NL63)[32]. Although the functional relevance of the two distinct D0 conformations (up versus down) remains to be established, the ability of the D0 to undergo such drastic conformational change is reminiscent of what was observed in the SARS-CoV-2[33-35], SARS-CoV[36] and MERS-CoV S proteins[37], in which the transition from the down to the up conformation of the respective RBD is obligatory for the receptor binding.

To better understand the structure-activity relationship of the G2 PEDV S in its native environment, we employed cryo-electron tomography (cryo-ET) subtomogram averaging (STA), and cryo-electron microscopy (cryo-EM) single-particle reconstruction (SPR) to characterize the structure and dynamics of the S protein of a highly virulent Pintung 52 (PT52) strain[38,39] of PEDV in situ on the intact viral particles. The PEDV PT52 was identified in Taiwan in 2013, and it is closely related to the CO/13 strain. We revealed distinct D0 up and down conformations of the PEDV PT52 S that can be asynchronously adopted by each of the three protomers of the S trimer. We further obtained recombinant S protein of the PT52 strain using the porcine IPEC-J2 intestinal epithelial cell line, which is the native host cell line for PEDV. The porcine cell line-derived S protein was analyzed by cryo-EM and mass spectrometry (MS) to elucidate the glycosylation patterns and their three-dimensional (3D) structures. Protein glycosylation plays a pivotal role in protein folding, pathogen-host recognition, and immunity evasion[40]. The integration of cryo-EM and MS to delineate the glycosylation patterns on glycoproteins, particularly for coronavirus S proteins, has emerged to become a powerful tool to help understand how individual glycans contribute to the biological functions[30,41-44]. While the cryo-EM structure of the IPEC-J2 cell line-derived S protein was identical to the recombinant S protein derived from the commonly used human HEK293 Freestyle (HEK293F) cell line, the glycosylation patterns differed in the amount of sialylation: the former sample exhibited more abundant sialylation. We further demonstrated by site-directed mutagenesis and cryo-EM analysis that the difference of a single N-glycan can significantly modulate the up/down conformation of the D0. The results provide atomic-level information that explains the immunity-escaping response of G2 PEDVs and offer hints to future research in PEDV and other alphacoronaviruses.

## Results

### Cryo-electron tomography and subtomogram averaging

To investigate the structural features of PEDV S in situ, we collected 90 tilt series of the intact PEDV PT52 particles by cryo-ET for subsequent STA analyses. Many S proteins were readily visible in the reconstituted tomograms (Fig. 1a). To generate an initial template model of the S protein, the tomograms were low-pass filtered to improve the signal-to-noise ratio, and the positions and orientations of 40 S proteins were manually defined. Subtomograms were extracted at the defined coordinates, followed by several iterations of STA with a C3 symmetry to yield a low-resolution template of the PEDV S for automated template matching to extract subtomograms corresponding to the S protein from within the intact viral particles. Knowing that the S proteins are located on the viral membrane surface, we devised a procedure to limit the template matching process within the proximity of the viral membrane instead of an unrestricted search process throughout the entire tomograms.

To achieve this goal, we first defined the coordinates of the viral membrane, which has a strong image contrast, as reference points to guide the automated search of the S proteins (Supplementary Fig. 1). The centers and radii of 462 viral particles were manually defined, which exhibited a bimodal distribution in their particle sizes (Supplementary Fig. 2a). An oversampled grid of coordinates with a 5 nm spacing was generated for each sphere that approximated the viral particle, and subtomograms were extracted from these coordinates. The viral membrane within the subtomograms was located by one iteration of STA. To approximate the location of the S proteins, the coordinates of the membrane bilayers were shifted 14 nm away from the centers of the viral particles, and a new set of subtomograms were extracted from these new coordinates, followed by several iterations of STA, during which the first iteration of STA aligned against the low-resolution initial template of the S proteins. Misaligned and duplicated subtomograms were iteratively removed by (i) imposing a particle distance threshold to remove subtomograms that were too close to each other, (ii) imposing a distance threshold with respect to the membrane to avoid clashing with the membranes or unrealistic departures from the membranes, (iii) applying a cross-correlation cut-off value to remove subtomograms that were too noisy, (iv) removing artifacts that arise from the high contrast ice-carbon boundaries, and (v) removing particles that have unrealistic tilt angle with respect to the membrane. Through these iterative filtering steps, we obtained 15,065 S protein subtomograms for further processing, corresponding to approximately 32 S proteins per virus.

### Conformational heterogeneity of the PEDV S within the D0

The additional D0 at the N-terminus of the S protein is a unique feature of alphacoronaviruses. Two independent studies on the homologous PEDV S reported two distinct conformation states, with one D0 being in the upward conformation (D0-up state)[12] and the other in the downward conformation (D0-down state)[31]. The D0-down state has the D0 tucked against the S2 subunit, while in the D0-

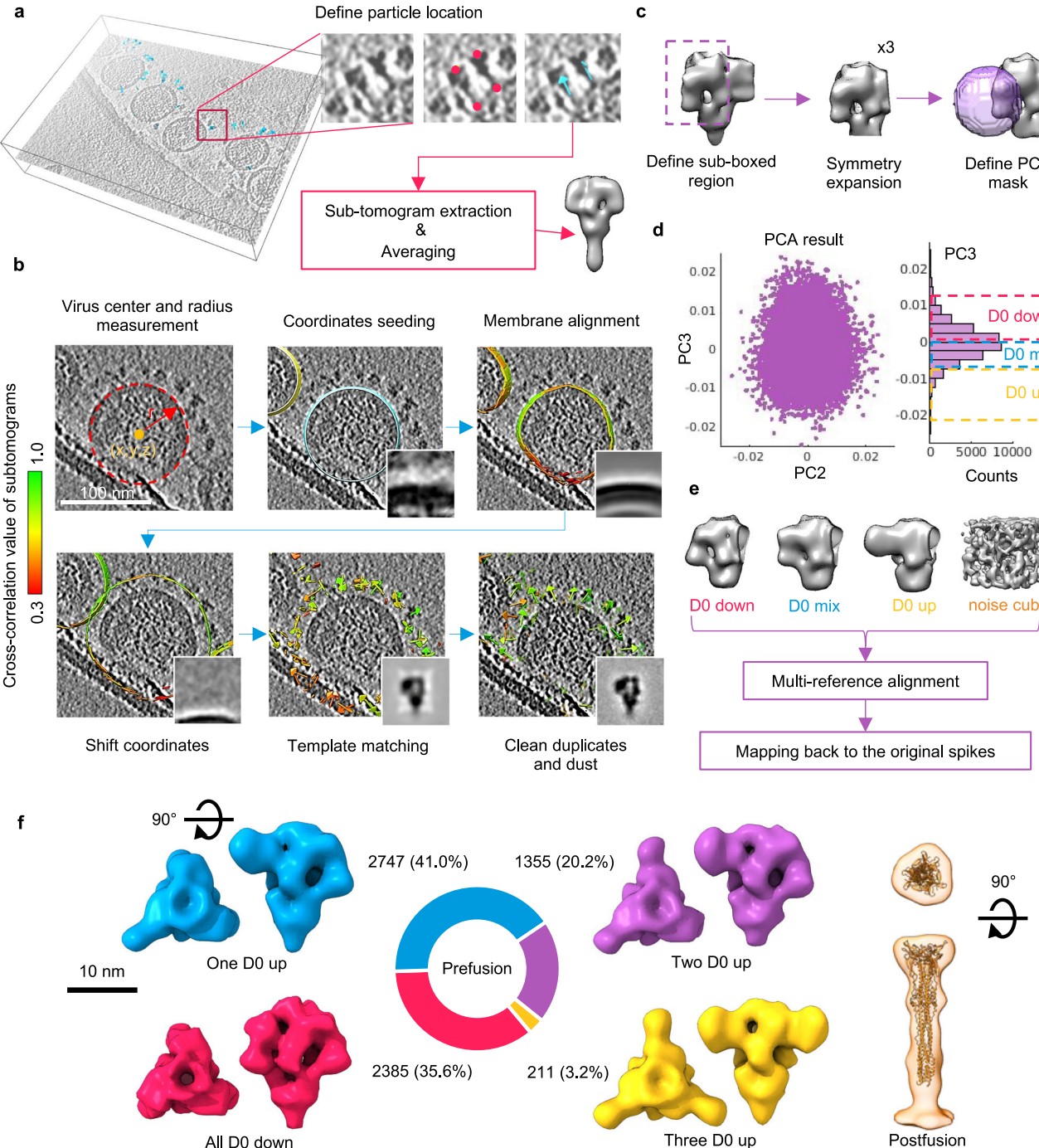

**Fig. 1 | Structural determination of the PEDV PT52 S by cryo-ET and STA.**
**a** Workflow of initial template generation for the S protein. Positions and orientations of S proteins were manually defined as indicated by fuchsia circles and cyan arrows, respectively, to extract the subtomograms. Iterations of STA were performed to generate the initial template. Indicated by hot pink circles and cyan arrows, respectively, to extract the subtomograms. Iterations of STA were performed to generate the initial template. **b** Workflow of subtomogram analysis. Membrane coordinates were approximated by a dashed red circle to define the center (x, y, z) and radius (r) of the viral particle. The membrane patches were aligned and colored green-to-red corresponding to high-to-low cross-correlation (CC) as indicated by the color scale bar on the left. The membrane coordinates were shifted outward from the center to help identify spikes. Duplicated spike coordinates and dust were removed before further analyses. **c** Symmetry expansion and mask definition for PCA analysis. Each S protein subtomograms were sub-boxed (dashed purple rectangle) to triplicate

particles by symmetry expansion. A spherical PCA mask colored in purple was defined to approximate the location where D0 heterogeneity was expected.
**d** Scatter plot of PC2 and PC3 of the PCA to reveal the D0 heterogeneity (left panel). Projection along PC3 was divided into three subclasses corresponding to D0 down, D0 up, and their mixture, i.e., D0 mix (right panel). **e** Averaged D0 conformation models (up, down, and mix) and a noise volume were used as reference templates in the multi-reference alignment task to classify all the protomer subtomograms. Classified protomers were mapped back to the original S trimers for further refinement. **f** Our cryo-ET results of PEDV PT52 S structures revealed that S protein with four different D0 conformational arrangements in the pre-fusion state and one in the post-fusion state. The pie chart illustrates the respective populations which include $D0_{DDD}$ (hot pink), $D0_{UDD}$ (blue), $D0_{UUD}$ (purple), and $D0_{UUU}$ (yellow). Orthogonal views of the reconstructed postfusion S structure in a semi-transparent orange surface superimposed with a homology model.

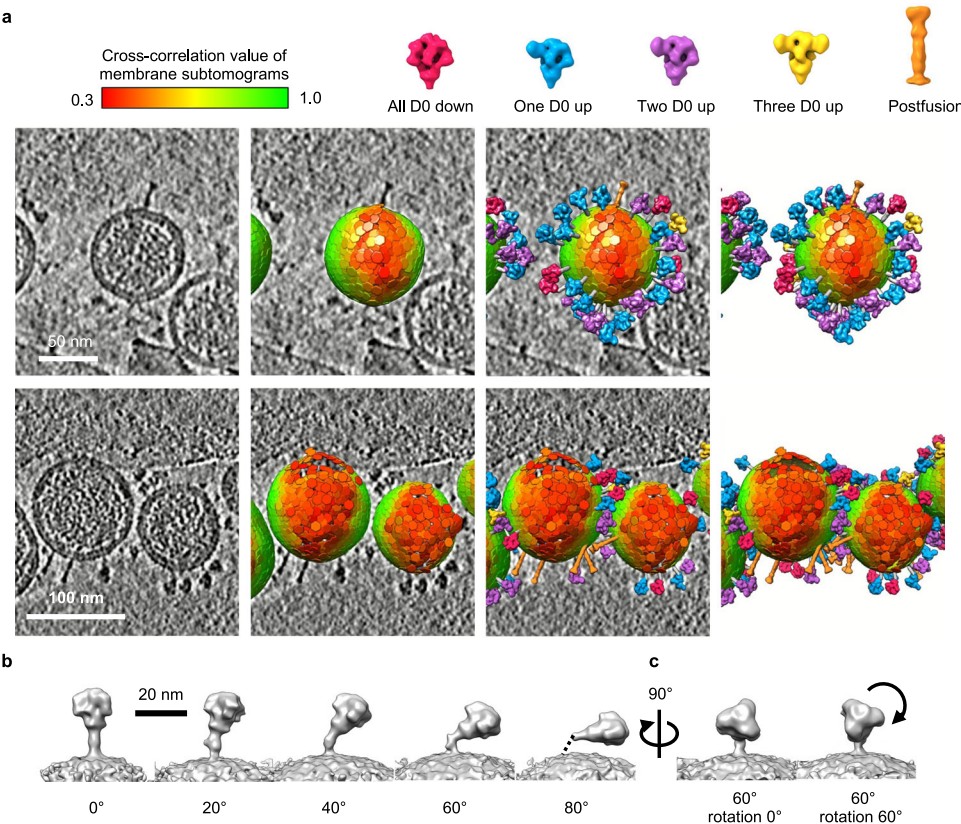

**Fig. 2 | Representative reconstructions of intact PEDV PT52 viral particles with different S protein structures. a** Representative reconstructions of the intact viral particles. The bottom row highlights the clustering of the postfusion S proteins. The reference map of S protein structures as defined in Fig. 1f. The membrane subtomograms were colored green-to-red corresponding to high-to-low CC values as indicated by the color scale bar on the top. **b** Subtomogram-averaged structures of the S proteins at 0, 20, 40, 60, 80° tilt. The stalk region of the 80° tilt S protein could not be resolved, which is indicated by a dashed line. **c** Selected snapshots of the S protein with a 60° tilt, illustrating the in-axis rotation with respect to its three-fold symmetry axis.

up state, the lifted D0 adopts a propeller-like conformation. It is important to note that the two previously reported structures are derived by imposing a C3 symmetry during the cryo-EM data processing; therefore the D0 of the three protomers can only adopt an all-up or all-down conformation. To better examine the structural features of the S protein of our PEDV PT52 sample, we performed a heterogeneity analysis of the D0 region of the S protein without imposing any symmetry constraints. First, we shifted the center of each S protein subtomogram to one of the three protomers. We repeated it to the other two protomers to enable particle sub-boxing to generate three smaller subtomograms corresponding to the three S protomers (Fig. 1c). Second, a principal component analysis (PCA) was performed with a focused mask created around the D0. Among the first few principal components, the third principal component (PC3) exhibited the most conformational variability of the D0. We, therefore, performed subtomogram averaging within different ranges of PC3 values to generate S protein protomers with distinct D0-up and D0-down conformations (Fig. 1d, e). We carried out a multi-reference alignment (MRA) to classify all the protomer subtomograms according to their D0 states. Four reference models were used for the MRA, namely D0-up, D0-down, D0-mix, and a noise cube (Fig. 1e). The D0-mix protomer model represented a protomer with a mixture of D0-up and D0-down states; the noise cube was a volume with random noise as a negative control for the classification. The MRA analysis yielded population distributions of 18.9%, 45.2%, 34.8%, and 1.1% for the D0-up, D0-down, D0-mix protomers, and noise, respectively. The minimal number of subtomograms falling into the noise category demonstrated the robustness of the template matching and filtering procedure.

By mapping all the annotated subtomograms of individual protomers to their trimeric assemblies without symmetry restrain, 35.6% of the S proteins adopted an all D0-down arrangement (D0$_{DDD}$), 41.0% of the S proteins adopted a one D0-up and two D0-down arrangement (D0$_{UDD}$), 20.2% of the S-proteins exhibited a two D0-up and one D0-down arrangement (D0$_{UUD}$), and 3.2% of the S proteins exhibited an all D0-up arrangement (D0$_{UUU}$). Any S protein that contains one or more D0-mix or noise cubes was excluded for further analyses. An independent half-set refinement was performed for all subsets of subtomograms. For the D0$_{DDD}$ and D0$_{UUU}$, a C3 symmetry was imposed in the subsequent refinement steps. Finally, the D0$_{DDD}$, D0$_{UDD}$, D0$_{UUD}$, and D0$_{UUU}$ structures were refined to a final resolution of 19, 27, 25, and 29 Å, respectively (Fig. 1f and Supplementary Fig. 2b).

### Distribution and structural flexibility of PEDV S

By mapping the individual S proteins in the prefusion state onto the virus particles according to their subtomogram coordinates, 33 ± 5 S proteins were identified on each virus regardless of the size of the virus particles (Supplementary Fig. 2c). Additionally, we observed rod-like structures on the viral surface that resembled the postfusion S proteins. As the number of the postfusion-like structures is limited, we manually picked the postfusion subtomograms ($n = 132$) and employed the STA procedure with a C3 symmetry constraint to obtain an EM map with a nominal resolution of 31 Å, which can be nicely superimposed with a homology model based on previously reported postfusion S protein structures of other CoVs (Fig. 1f)[45,46]. However, not all viruses contained the postfusion-like structure, and the postfusion structures tended to cluster (Fig. 2a and Supplementary Movie 1). The formation of the postfusion state

could be attributed to the addition of trypsin in the cell culture medium during the viral particle production. A small amount of trypsin in the cell culture medium is required for efficient S1/S2 cleavage and propagation of PEDV[47,48].

Geometrical analysis of the prefusion S proteins over the viral surface showed a peak distribution of the tilt angle of the S protein with respect to the membrane normal at around 80° (Supplementary Fig. 2d). A comparison of the subtomogram averaging of the S protein as a function of the tilt angle showed a less resolved stalk density for the S proteins with higher tilt angles, indicating a higher degree of conformational heterogeneity (Fig. 2b). In-axis rotation of the S protein around the long principal axis of the S protein was also observed (Fig. 2c and Supplementary Movie 2).

### Details of the PEDV S structure and its rearrangement uncovered by Cryo-EM single particle reconstruction

To obtain further insights into the structure and dynamics of the PEDV S at a higher resolution, we collected cryo-EM data from the same batch of viral particles. With a higher electron exposure for the cryo-EM data collection, the S proteins on the viral particles were readily visible in the raw micrographs (Fig. 3a). Due to the strong signals of the viral membrane that dominated the image alignment, we skipped the conventional 2D classification step and resorted to 3D classification to remove bad particle images. The resulting initial 3D reconstruction displayed a well-defined trimeric S protein structure with clearly separated central helices. The particles were refined with a C3 symmetry followed by symmetry expansion to extract further information with regard to the local motions within individual protomers. In line with our cryo-ET analysis, the 3D classification at the level of individual protomers revealed the existence of at least two distinct conformations that differed in the relative position of the D0, i.e., D0-up versus D0-down (Supplementary Fig. 3). The set of S trimer particles with all three protomers in the D0-down conformation was refined with C3 symmetry, resulting in a $D0_{DDD}$ map with a global resolution of 4.7 Å (Fig. 3b and Supplementary Fig. 3). 68% of the final S trimer particles corresponded to the $D0_{DDD}$ arrangement. The refinement of the S trimer particles containing D0-up protomer particles yielded a $D0_{UDD}$ map with a global resolution of 6.4 Å (Fig. 3c and Supplementary Fig. 3). Further 3D variability analysis[49] of the S trimer particles with the D0-up protomers revealed three distinct classes corresponding to the $D0_{UDD}$ arrangement (50%), the $D0_{UUD}$ arrangement (48%), and the $D0_{UUU}$ arrangement (2%; Supplementary Fig. 4a–c). The underrepresentation of the D0-up population in the cryo-EM analysis compared to that observed by cryo-ET may be attributed to the stringent particle image filtering criteria used in cryo-EM, which removed around 70% of the initial images to prioritize high-resolution information.

The atomic models corresponding to the PEDV S in the $D0_{DDD}$ and $D0_{UDD}$ arrangements were built (Fig. 3d, e). Superimposition of the D0-down and D0-up S protomer models revealed a rigid body motion of the D0 with an overall fold resembling the previously reported PEDV structures (Supplementary Fig. 4d, e). To describe the D0 motion between the D0-up and -down states, we defined **k**, a vector originating from the hinge connecting the D0 and S1-NTD to the center of mass of the D0. Going from the D0-down to D0-up conformation, D0 experienced a 136° in-axis rotation about **k**, and **k** underwent a 93° rotation away from the S2 subunit pivoting at the hinge between the D0 and NTD (Fig. 3g).

### Insight into glycosylation of the PEDV S by integrative structural approach

Despite the moderate resolutions of our cryo-EM maps, we observed well-defined secondary structure elements and bulky side chains, with clear protrusions from the asparagine side chains at 21 out of 23 putative N-glycosylation sites, which we attributed as the glycan structures (Fig. 4, and Supplementary Table 1). In most cases, at least two N-acetylglucosamine (GlcNAc) moieties common in all N-glycoforms could be modeled[40] (Fig. 4e–h). In particular, the N-glycan at Asn324 was positioned at the hinge region that connects the D0, the EM map of which could be resolved in both the D0-up and D0-down conformations (Fig. 4a, b) thereby allowing atomic model building for the first two GlcNAc moieties (Fig. 4g). We postulate that this particular N-glycan could potentially generate steric hindrance to the interconversion of the D0 between the up and down states (Fig. 3g).

To better define the glycoforms of the individual N-glycans, we performed MS-based glycopeptide analysis to determine the glycoforms at individual N-glycosylation sites for model building[30,41]. Our initial attempt to obtain PEDV PT52 viral particles from six infected piglets failed to yield a sufficient quantity of samples for MS analysis. Instead, we succeeded in developing the porcine IPEC-J2 intestinal epithelial cell[50], which is the native host cell line for PEDV, to generate a stable cell line to enable the production of PEDV PT52 S[50], used here to generate recombinant protein for detailed structural and MS characterizations (Fig. 4i, Supplementary Fig. 5, and Supplementary Table 1). The glycoforms of 18 of the 28 predicted N-glycosylation sites on PEDV PT52 S were quantitatively determined, of which three sites were predominantly high-mannose type glycans, and 13 were predominantly complex-type glycans. Additionally, several N-glycans, namely N62, N118, N216, N300, N344, N351, N556, and N667, contained terminal Hex-HexNAc units capped by an extra Hex residue, corresponding to the Galα1-3 Gal epitope, which is absent in human (Fig. 4i and Supplementary Table 1)[50–52]. In addition to the core region of the N-glycans that can be reliably built into the observed cryo-EM maps, we could build a glycan chain of up to six sugar moieties for the most extended glycan density at Asn216 in the $D0_{UDD}$ map (Fig. 4f). Of the 11 complex-type N-glycans, the core fucose moiety could be modeled to fit the additional densities that extended from the side-chains of Asn216, Asn300, Asn556, and Asn743 (Fig. 4e, f). On the one hand, the cryo-EM maps did not show additional density at Asn62 and Asn1196 that harbored complex-type N-glycans according to the MS analysis. On the other hand, the cryo-EM maps showed clear protrusions at six sequons – Asn324, Asn514, Asn425, Asn726, Asn781, and Asn787 – that could not be observed by MS analysis (Supplementary Table 1). These results, therefore, underlined the importance of the integrated approach for glycosylation analyses (Supplementary Fig. 5).

To fill in the gap of the missing glycoform information, particularly for Asn324, we produced recombinant PEDV PT52 S using the HEK293F cells and repeated the same MS glycopeptide analysis. We identified 21 of the 28 expected N-glycans, but that of Asn324 remained undetected (Supplementary Fig. 6, Supplementary Fig. 7b and Supplementary Table 1). As an alternative solution, we produced a recombinant PEDV S from a closely related HC070225 strain[26], which shared a 93.7% sequence identity with PEDV PT52 S. The two PEDV strains shared the same sequons for the 24 putative N-glycosylation sites, of which four have different but synonymous amino acid sequences. In addition, the HC070225 strain contains an additional N-glycosylation sequon at Asn723 that was absent in PEDV PT52 S due to a sequence variation (Supplementary Fig. 7c and Supplementary Table 1). The glycoforms of 21 out of the 29 predicated N-glycosylation sites on the PEDV HC070225 S were quantitatively determined, including that of Asn325, which is equivalent to that of Asn324 of PEDV PT52 S. The major glycoform of Asn325 of PEDV PT52 S is a high mannose type with eight mannose residues (M8), which could be attributed to the spatial occlusion by the D0 and the neighboring protein structures that prevent further processing into complex type glycoforms[42].

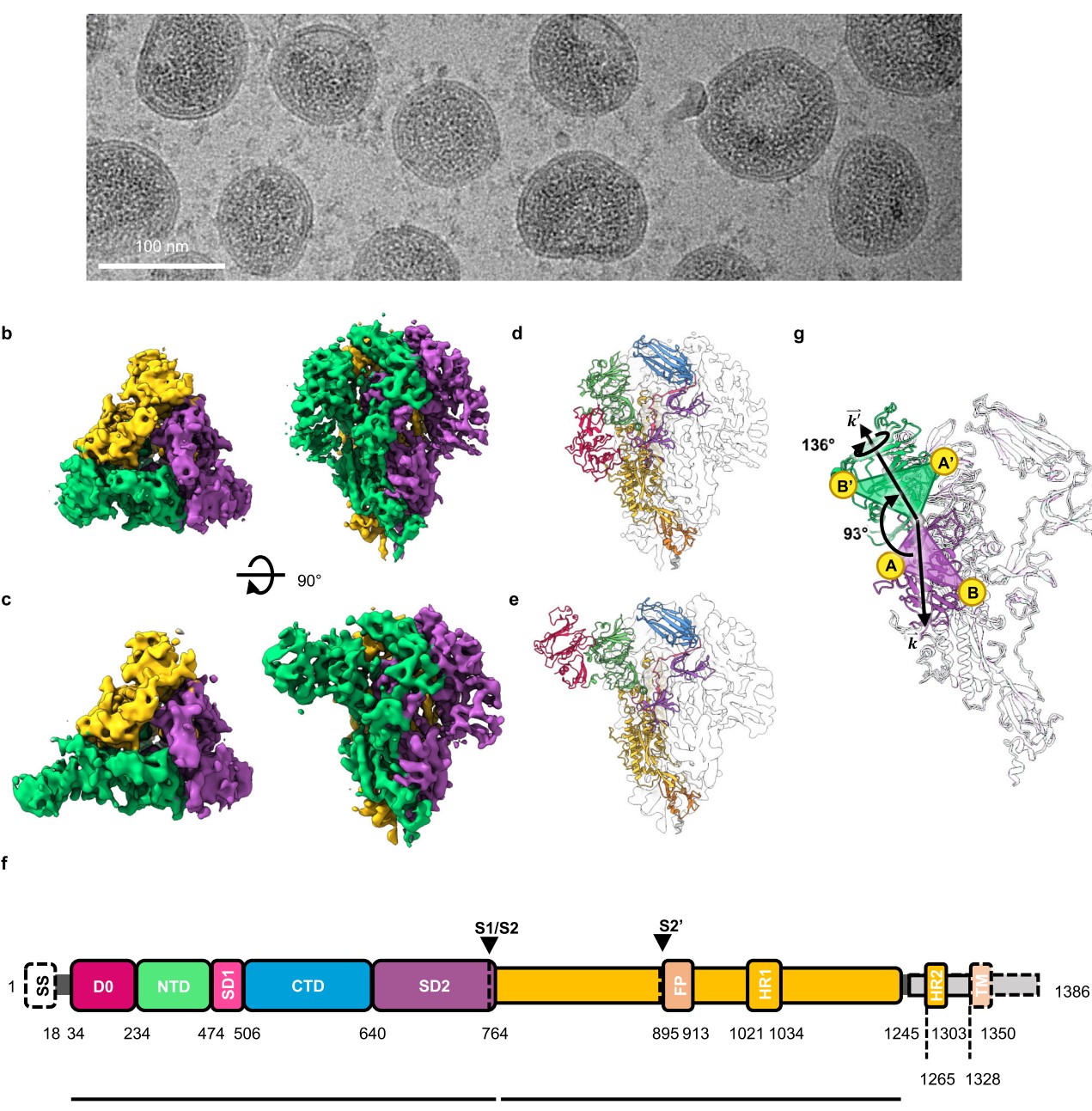

**Fig. 3 | In situ cryo-EM analysis of the structure of PEDV PT52 S on intact viruses. a** Raw micrograph illustrating clear densities of S proteins on the viral membrane surface. 3701 micrographs were collected for this dataset (Supplementary Fig. 3). Orthogonal views of the cryo-EM maps of the D0-down conformation (D0$_{DDD}$) (**b**) and the D0-up conformation (D0$_{UDD}$) (**c**) with their respective atomic models are shown in cartoon representations in **d** and **e**. **f** Domain definitions of the PEDV PT52 S with the domain boundaries are indicated below. The individual domains are colored in the same scheme as in **d** and **e**. The

regions that are missing in the cryo-EM structures are outlined by dashed lines. SS, signal sequence; D0, domain 0; NTD, N-terminal domain of S1; SD1, subdomain 1 of S1; CTD, C-terminal domain of S1; SD2, subdomain 2 of S1; FP, fusion peptide; HR1, heptad repeat 1; HR2, heptad repeat 2; TM, transmembrane domain; The black arrows highlight the protease cleavage sites S1/S2 and S2'. **g** Superimposition of the D0-down (purple) and D0-up (green) conformation S protomer models depicting the motion of the D0.

## Putative receptor binding domain conformational changes in recombinant PEDV PT52 S

Having obtained the recombinant PEDV PT52 S from the IPEC-J2 cells, we subsequently determined its cryo-EM structure to a nominal resolution of 3.1 Å without imposing a symmetry constraint (Fig. 5 and Supplementary Fig. 8). The structure exhibited a D0$_{UUU}$ arrangement as opposed to the predominantly D0$_{DDD}$ arrangement for the intact

virion-derived S protein structure. Through 3D classification, we also observed a minor conformation that adopted a D0$_{UUD}$ arrangement (Fig. 5c, d). In addition to the up/down transition of the D0, we observed a major conformational change in the C-terminal domain (CTD) that resembles the conformational changes of the RBDs of SARS-CoV-2 S[33–35,53–58], SARAS-CoV and MERS-CoV[59]. Two of the three CTDs underwent an 82° elevation from the closed conformation to

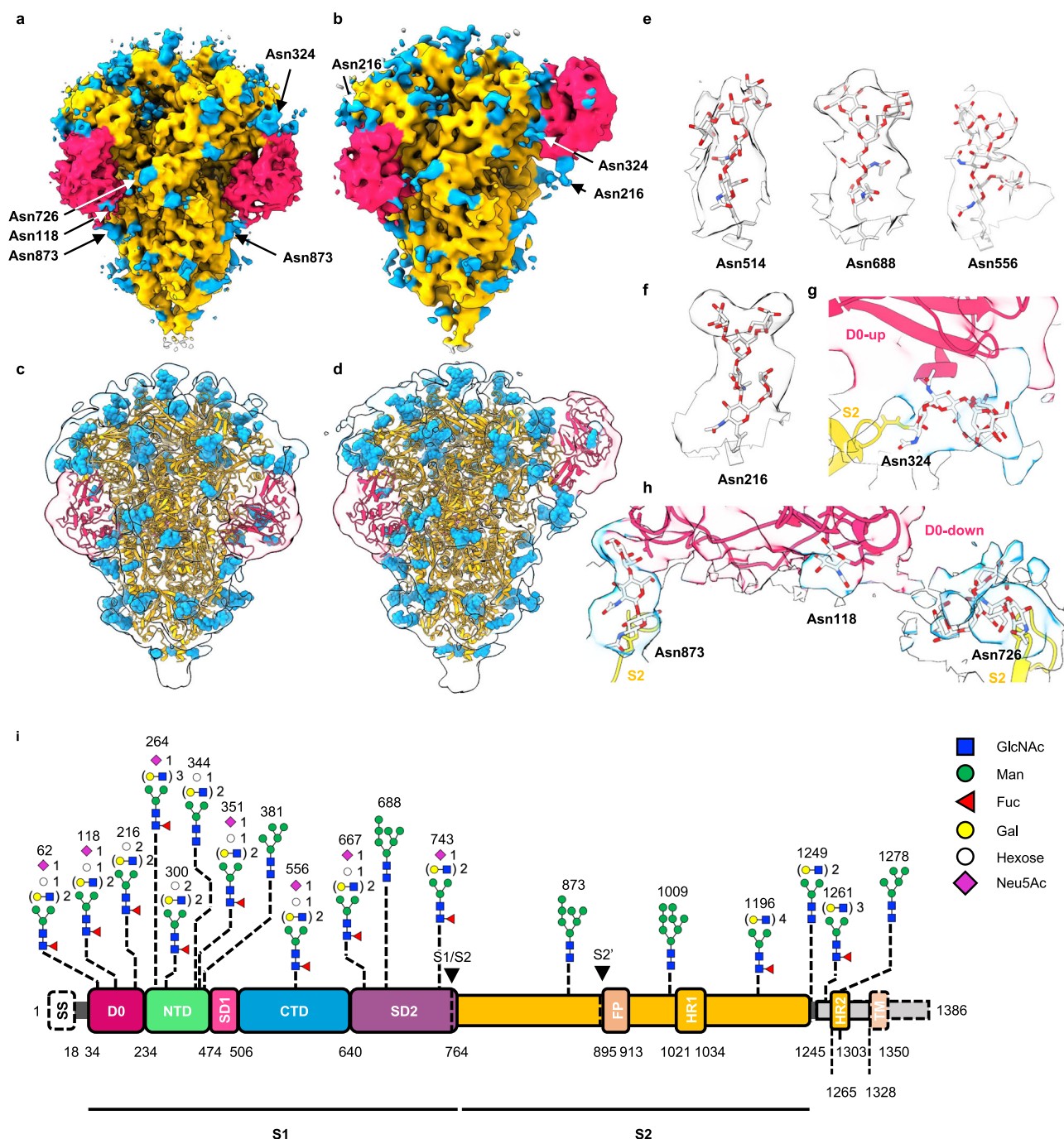

**Fig. 4 | Glycosylation analysis of the PEDV PT52 S.** The cryo-EM maps of PEDV PT52 S in all D0-down (D0$_{DDD}$) arrangements (**a**) and only one D0-up (D0$_{UDD}$) arrangement (**b**). **c**, **d** The EM maps were displayed at a low threshold with the D0 and protruding glycan densities highlighted in hot pink and blue, respectively. The respective S protein cryo-EM maps were low-pass filtered, and shown as transparent isosurfaces superimposed with their respective atomic models. **e** Cryo-EM densities of representative glycans and their modeled atomic coordinates. **f** EM map and atomic model of the N-glycan on Asn216. **g** N-glycan on Asn324 of the S2 subunit that makes extensive contact with D0 in the D0-up conformation.

**h** Protein-glycan interactions between the D0 and S2 subunit in the D0-down conformation. **i** The representative glycoforms of the recombinant PEDV PT52 S expressed in IPEC-J2 cells. The individual glycans are depicted in symbols following the standard Symbol Nomenclature For Glycans (SNFG)[88], as indicated on the right. Note that the complex type N-glycans of PT52 S produced in IPEC-J2 cells carried extra hexoses (open circle) attributable to terminal Galα1-3Gal-GlcNAc, in addition to Neu5Ac (magenta diamond)-sialylated Gal-GlcNAc antenna, at Asn62, Asn118, Asn216, Asn300, Asn344, Asn351, Asn556, and Asn667.

adopt an open conformation (Fig. 5e–g) that was not observed in the structure derived from intact viral particles (Fig. 3). Such a domain motion is not observed in the previously reported cryo-EM structures of the other PEDV S proteins[12,31]. The D0-up conformation of the recombinant PEDV PT52 S also differed slightly from that derived from intact viral particles (Supplementary Fig. 9). While the identity of the PEDV receptor is currently unknown, the CTD motion observed in the

recombinant PEDV PT52 S strongly suggests its involvement in host receptor binding.

To evaluate the effect of the N-glycan at Asn324 in modulating the D0 conformation of PEDV PT52 S, we used site-directed mutagenesis to replace Thr326 into isoleucine (T326I) according to the sequence alignment of PT52, CO/13 and CV777; CO/13 has isoleucine at the equivalent position, leading to the loss of the N-glycosylation sequon,

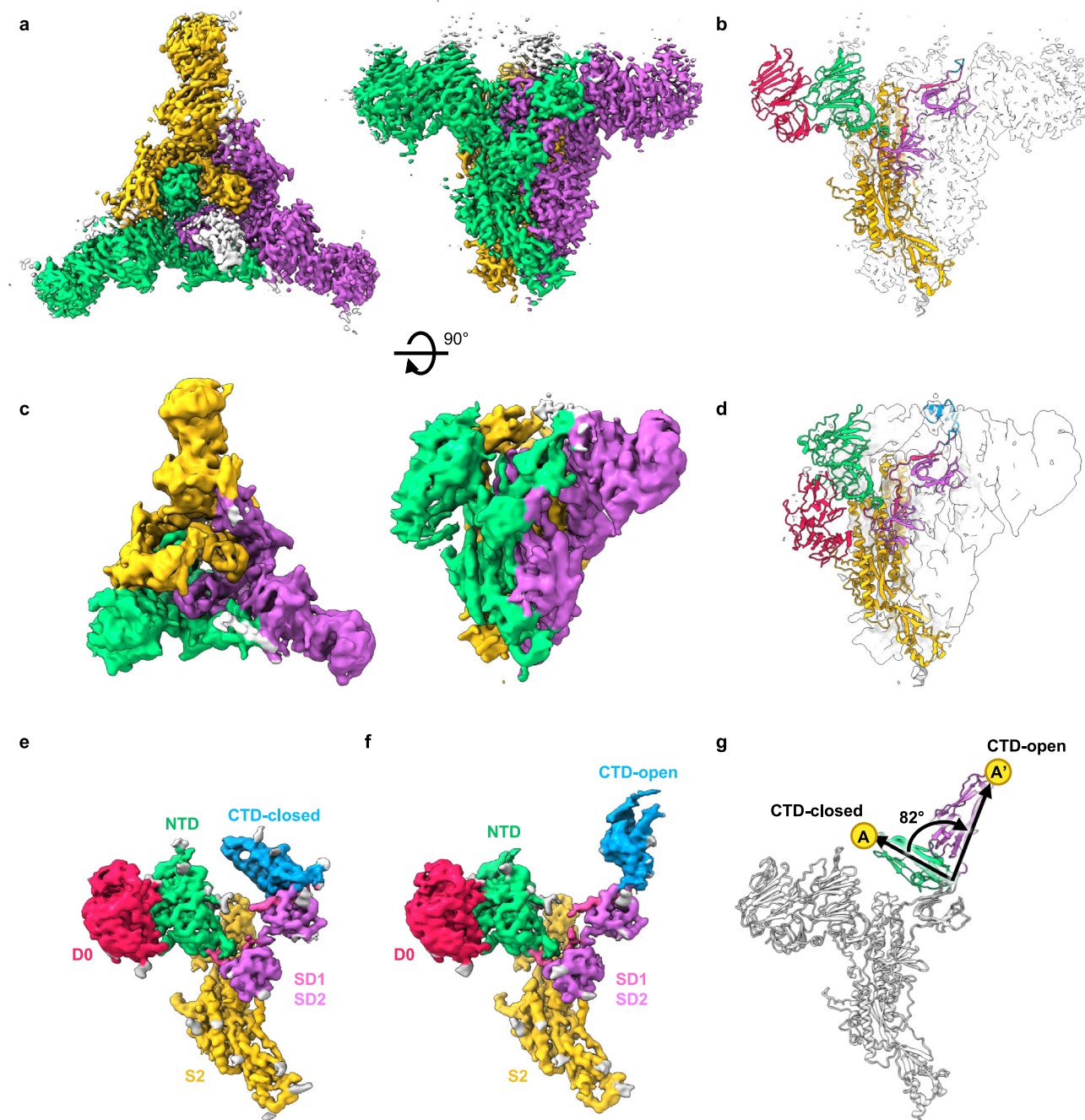

**Fig. 5 | Cryo-EM analysis of the structure of recombinant PEDV PT52 S derived from IPEC-J2 cells.** Orthogonal views of the cryo-EM maps of the D0$_{UUU}$ arrangement (**a**) and the D0$_{UUD}$ arrangement (**c**) with their respective atomic models are shown in cartoon representations in **b** and **d**, respectively. The individual domains are colored following the same scheme as in Fig. 3. Refined maps of D0-up protomers with CTD in closed (**e**) and open conformation (**f**) colored by individual domains. **g** Superimposition of the CTD-close (green) and CTD-open (purple) conformation S protomer models depicting the motion of the CTD.

NXS/T (Supplementary Fig. 10e). We used HEK293F cells to produce the recombinant PEDV PT52 S with and without the T326I mutation to determine their cryo-EM structures (Supplementary Figs. 11, 12). Despite the variations in the glycoforms between the IPEC-J2 cell- and HEK293F cell-derived PEDV PT52 S, the atomic structure of IPEC-J2 cell-derived PEDV PT52 S perfectly fits to the most representative (82% of the total population) cryo-EM map of HEK293F cell-derived PEDV PT52 S, which adopted a D0$_{UUU}$ arrangement with a lower nominal resolution of 4.5 Å due to preferred orientation issues and smaller data size (Supplementary Fig. 13). Two additional classes of EM maps were also deduced from the cryo-EM data, which corresponded to the D0$_{UUD}$ and D0$_{DDD}$ arrangements, corresponding to 7%, and 11% of the total

population, respectively (Supplementary Fig. 14). The T326I variant also exhibited a D0$_{UUU}$ arrangement whose EM map could fit to the IPEC-J2 cell-derived PEDV PT52 S atomic model, but the relative population was much lower (23%). Instead, the T326I variant populated more D0-down conformation with the 19%, 16% and 42% of the population being in the D0$_{UUD}$, D0$_{UDD}$ and D0$_{DDD}$ arrangements, respectively (Supplementary Fig. 14). Quantification of the relative D0-up vs D0-down conformations at a level of individual protomers of the S trimers showed the same trend (Supplementary Fig. 14e). In other words, the loss of the N-glycan at Asn324 due to the T326I mutation resulted in an almost four-fold increase of the D0-down conformation for the HEK293F cell-derived recombinant PEDV PT52 S. This finding

confirmed our hypothesis that this N-glycan is promoting the D0-up conformation (Supplementary Fig. 14).

## Discussion

Two structures of PEDV S variants have been reported by two independent studies in which one adopts a D0_UUU arrangement[12] and the other a D0_DDD arrangement[31]. Both studies imposed a C3 symmetry on the final cryo-EM structure such that all three protomers could only adopt either all D0-up or all D0-down conformations. Nevertheless, these studies reveal the possibility of the D0 domain to undergo large conformational changes that have not been reported for S proteins of other alphacoronaviruses. The S proteins of HCoV-NL63 and HCoV-2293 are in the all D0-down conformations while that of a feline CoV, FIPV, is in the all D0-up conformation[30,32,60]. In this study, we combined cryo-ET and cryo-EM to investigate the conformational landscape of the PEDV PT52 S in situ on intact viral particles. The in situ structural analyses have been extensively used to study SARS-CoV-2, enabling the identification of an opening and closing transition of the RBD[61-63]. The in situ approach was also used to reveal the structure of HCoV-NL63 S by cryo-EM[32]. In this case, however, the analysis yielded a single conformation of the protein that corresponds to D0_DDD arrangement[32]. The structural analysis of the PEDV PT52 S on intact viral particles presented here enabled the identification of the co-existing D0-up and D0-down conformations of the S protein, demonstrating heterogeneity among alphacoronaviruses. Importantly, we showed that each of the three protomers of the PEDV S could independently adopt either conformation in an asynchronous manner. Despite the discrepancy between cryo-ET and cryo-EM in terms of the absolute proportions of the individual states, it is clear that the D0-down conformation is energetically more favorable than the D0-up conformation (Fig. 1f and Supplementary Fig. 4a–c).

Our cryo-ET analysis showed that 30.3% of the protomers are in the D0-up conformation (Fig. 1f). Assuming that each of the three protomers can independently switch between the D0-up and D0-down conformation, the overall probability p of finding n protomers in the D0-up conformation in an S trimer can be expressed in the following equation (Eq. 1)

$$p(n) = C_n^3 q^n (1-q)^{(3-n)} \qquad (1)$$

where $q$ is the probability that a given protomer adopts a D0-up conformation, which is 30.3%. The results of the above theoretical calculation are 33.9%, 44.2%, 19.2%, and 2.8% for D0_DDD ($n = 0$), D0_UDD ($n = 1$), D0_UUD ($n = 2$), and D0_UUU ($n = 3$), respectively. The experimental findings derived from the cryo-ET analysis (Fig. 1f) correlate exceptionally well with the theoretical prediction, with a correlation $R^2$ of 0.99 (Supplementary Fig. 15). All aforementioned PEDV S arrangements were identified in the cryo-EM dataset (Supplementary Fig. 4), but the D0-up populations are underrepresented potentially due to the stringent filtering criterion of the particle images that are prioritized for high-resolution information, which is not as strictly imposed for the cryo-ET analysis.

What could be the molecular basis of modulating the up/down transition of the D0 of the PEDV S? Other than the differences in the sample preparation or the data processing, the discrepancy between our findings and the previous studies[12,31] could be attributed to the differences in the S protein sequences. The PEDV CV777 strain, in which only the D0-up conformation is observed, belongs to the G1 PEDV class[12]. The PEDV CO/13 strain, in which only the D0-down protomer is observed[31], and the PEDV PT52 strain, which we used for the structural analysis herein, belong to the more recently evolved G2 PEDV class[38,39,64]. The sequence alignment revealed that while the S proteins of PEDV PT52 and CO/13 share >99% sequence identity in the S1 and S2 subunits; they share a considerably lower (<94%) sequence identity with PEDV CV777. The sequence variations between the G1 and

G2 PEDVs mentioned herein are mostly in the S1 subunit (Supplementary Table 2). Indeed, the structure-based alignment of the CV777, CO/13, and our PT52 S protomer models revealed that the central helices of the S2 subunit are the most structurally conserved, while the S1 subunit is the most variable (Supplementary Fig. 16). Recent years have witnessed an exponential growth of reported coronaviruses S protein structures. A large number of variants of SARS-CoV-2 S structures illustrate how a few amino acid differences can significantly alter the relative up and down populations of the S protein RBD[33-35,53-58]. The observed D0 heterogeneity in PEDV S could be another example of this.

Previous studies on SARS-CoV-2 and FIPV S showed that glycosylation could affect the domain motions[30,65]. We therefore investigated the impact of sequence variation in the context of N-glycosylation on PEDV S variants. Although MS analysis did not identify the glycopeptide corresponding to the N-glycan at Asn324 (Fig. 4i), the cryo-EM map of PEDV PT52 S derived from intact viral particles showed clear evidence of N-glycosylation at this position that is located at the hinge region connecting the D0. The N-glycan at Asn324 (Asn321 in PEDV CV777 S) makes extensive contacts with the D0 in its up conformation in both the PEDV PT52 and PEDV CV777 (Fig. 4g and Supplementary Fig. 10), suggesting a structural role of this N-glycan in stabilizing the D0-up conformation. In contrast, the D0-down conformation exhibited a cluster of N-glycans on Asn726, Asn873, and Asn118, which made extensive contact with the D0 and S2 subunit (Fig. 4h). We used HEK293 cells to produce recombinant PEDV PT52 S with and without the T326I mutation, which will abolish the sequon for the N-glycosylation at Asn324 and Asn321 for PEDV PT52 S and PEDV CV777 S, respectively (Supplementary Fig. 10), reminiscent to the Thr-to-Arg mutation at residue position 19 around the hyper-antigenic supersite of the NTD of SARS-CoV-2 S found in the Delta variant[35]. As expected, the loss of the N-glycan at Asn324 led to a more populated D0-down conformation, suggesting that the degree of D0-up conformation in PEDV S may positively correlate with the virulence of the PEDV strains. Note that the recombinant PEDV PT52 S expressed in the HEK293F cells showed a slightly different glycosylation pattern to that observed in the recombinant PEDV PT52 S expressed in the IPEC-J2 cells, particularly in the extent of terminal sialylation, since the IPEC-J2 cells would generate competing terminal Galα1-3 Gal units that are absent in the HEK293F cell-derived recombinant protein (Fig. 4i, Supplementary Fig. 7 and Supplementary Table 1).

In the present study, the co-existence of up and down conformations of the D0 of PEDV PT52 S protein was revealed on intact viral particles. The up/down motion of the D0 may conceal the neutralizing epitopes on the S protein, thereby helping the virus evade host immune responses. Indeed, structural mapping of the recently reported antibody E10E neutralizing epitope of a G2 PEDV[26] revealed the sequestering of the E10E epitope in the D0-down conformation (Supplementary Fig. 17); the majority (70%) of the PEDV PT52 S protomer is in the D0-down conformation. Our experimental strategy may also serve as a blueprint for future studies on the structural polymorphism of S proteins and their functional implications in the context of host immune responses in intact viral particles that preserve most of the native environment. Another key finding is the up/down motion of the CTD that was only observed in the recombinant PEDV PT52 S but not in the intact viral particles. The lack of clear CTD motion in the cryo-ET and cryo-EM data of intact viral particles implies that such a motion may be an artifact due to the detachment of the S protein from the viral membrane. However, the CTD motion is not observed in the previously reported PEDV S structures, which are also based on isolated recombinant S proteins[12,31]. Nevertheless, the well-defined CTD motion in the recombinant PEDV PT52 S, which closely resembles the RBD motion observed in SARS-CoV-2 S[33-35,53-58], is strong evidence of its potential implication in host receptor binding.

In summary, our study sheds light on the conformational variability of the virulent PEDV PT52 S protein and provides a possible molecular mechanism controlling the D0 motion through N-glycan–protein interactions. We postulate that the D0-up and D0-down transition may contribute to the immune evasion of G2 PEDVs. It remains to be established as to how the D0 motion could contribute to the host-cell recognition, which requires the knowledge about the PEDV receptors, and the RBD of PEDV S. The conformational variability of the D0 and CTD reported herein offers a foundation for understanding the functional dynamics of the PEDV S, which may be applicable to other alphacoronavirus S proteins. The PEDV S structure presented in this study illustrates the conformational heterogeneity that represents a spectrum of possible permutations of the D0-up and D0-down conformations in the same sample. The ability to determine the conformational landscape in situ provides more physiologically relevant insights as opposed to the information derived from recombinant S proteins. As the D0 moves as a rigid body without obvious changes in the structure to the rest of the protomer, we suggest that the conformation adopted by one of the protomers of the PEDV S trimer maybe independent of the conformations of the adjacent protomer. The extensive protein-glycan interactions may be a contributing factor governing the conformational transition of the D0, which has potential implications in host immunity recognition.

## Methods

### Cells culture

The Vero C1008 cells (ATCC No. CRL-1586), the HEK293 cells (ATCC No. CRL-1573), and the Intestinal porcine epithelial cell line-J2 (IPEC-J2) cells (ACC701; Leibniz Institute DSMZ – German Collection of Microorganisms and Cell Cultures, Braunschweig, Germany) were cultured in the Dulbecco's modified Eagle's medium (DMEM, Gibco, NY, USA) supplemented with 10% fetal bovine serum (Gibco, Thermo Fisher Scientific, MA, USA) and antibiotic-antimycotic (Gibco, Thermo Fisher Scientific, MA, USA). The HEK293F cells (Invitrogen, CA, USA) were cultured in FreeStyle™ 293 Expression Medium (Gibco, Thermo Fisher Scientific, MA, USA).

### Virus propagation and inactivation

The PEDV-Pintung 52 (PT52), which belongs to the genotype IIb (G2b) PEDV, was isolated and passaged in the Vero cells with TPA medium, a DMEM-based medium (Gibco, Thermo Fisher Scientific, MA, USA) supplemented with 0.3% tryptose phosphate broth (Sigma-Aldrich, MO, USA), 0.02% yeast extract (Acumedia, CA, USA), and 10 µg/mL trypsin (Gibco, Thermo Fisher Scientific, MA, USA). After serial passages, the highly passaged PEDV PT52 passage 96 (PEDV PT52-p96, Genebank No. KY929406.1) with a titer of $10^6$ $TCID_{50}$/mL was obtained. To prepare the viral stocks, the PEDV PT52-p96 (defined herein as PT52) was diluted in TPA medium at a ratio of 1:1000 and subsequently propagated in the Vero cells with a 90% confluency. The supernatant was directly collected without freezing as more than 80% of cells showed visible cytopathic effects. The soluble phase was collected after centrifugation at 1620 × g for 20 min followed by filtration by using a 0.22 µm pore size Bottle Top Vacuum Filter (Corning, NY, USA). The viruses were fixed and inactivated by adding formaldehyde to a final concentration of 2% and incubating at 37 °C with regular shaking for one hour. The viral particles were pelleted by one-step sucrose cushion ultra-centrifugation padding with 20% sucrose (Sigma-Aldrich), and centrifuged (Beckman Coulter, CA, USA) at 72,411 × g with JA-25.50 rotor under 10 °C for 2.5 h. The viral pellets were re-suspended in the TNE buffer containing 50 mM Tris-HCl (pH 7.4, Sigma-Aldrich, MO, USA), 100 mM NaCl (Sigma-Aldrich, MO, USA), and 0.1 mM EDTA (Sigma-Aldrich, MO, USA). Complete inactivation of the virus was confirmed by rechallenging Vero cells.

### Virus purification

The viral pellet resuspensions were layered over a 10–60% linear sucrose gradient produced by the Gradient Master (BioComp Instruments, Fredericton, Canada) and ultra-centrifuged in an SW-55 Ti rotor (Beckman Coulter) at 193,911 × g for 1.5 h at 4 °C. The purified viral particles were collected from 40 to 45% interface, where an opaque virus band was observed. The sucrose in the buffer was removed by repeating washing steps on a 0.5 mL 100 kDa molecular weight cut-off (MWCO) column (Amicon, Merck Millipore, MA, USA) with TNE buffer. In total, 5 mL of TNE buffer (10 X column volume) was used. The purified viral particles were collected and stored at 4 °C for in situ cryo-ET/EM.

### Molecular cloning

The constructs of both PEDV PT52-passage 5 (PT52-p5, Genebank No. KY929405.1), and the Taiwan historic G1 strain HC070225 (Genebank No. KP768390.1) were described previously[66,67]. The open reading frame of PEDV Spike referring to PEDV PT52 was first introduced with double proline mutations ($^{1076}IL^{1077} \rightarrow {}^{1076}PP^{1077}$) to stabilize the spike in the prefusion state. The cDNA sequence corresponding to the extracellular domain of the PEDV PT52, was subsequently codon-optimized and synthesized by Genscript Corporation (Piscataway, NJ, USA). This gene was inserted into pcDNA3.4-TOPO (Invitrogen) which contained a trimerization domain (foldon) of T4 phage fibritin followed by c-Myc and a His-tag at the C-terminus. A glycosylation sequon mutation T326I was generated by using site-direct mutagenesis with two complementary primers (Mission Biotech, Taiwan):

Forward, 5'–CGCTTTAATATCAACGATATCTCTGTGATCCTGGCAGAG–3'

Reverse, 5'–CTCTGCCAGGATCACAGAGATATCGTTGATATTAAAGCG–3'

### Recombinant protein production

By using PolyJet™ In Vitro Transfection Reagent (SignaGen® Laboratories, MD, USA), the recombinant PEDV S proteins of PEDV PT52-p5, and the Taiwan historic G1 strain HC070225 were transiently expressed in HEK293 cells. In contrast, wild-type (WT) PT52 was stably expressed in IPEC-J2 cells by culturing the cells in fresh DMEM (Gibco, Thermo Fisher Scientific, MA, USA) supplemented with 10% fetal bovine serum (FBS, Gibco, Thermo Fisher Scientific, MA, USA) and 750 µg/mL of selective antibiotic Geneticin (G418; Gibco, Thermo Fisher Scientific, MA, USA) for two weeks to establish a stable cell line. The recombinant PEDV S proteins of WT PEDV PT52 derived from IPEDC-J2 cells were expressed in two different types of culture media, namely Freestyle 293 expression medium and DMEM with 10% FBS, containing 750 µg/ml Geneticin, and cultured for five days prior to harvest. For recombinant PEDV S proteins of WT PT52 and the T326I variant, their encoding plasmids were introduced into HEK293F cells (Invitrogen, CA, USA) individually by transient transfection with polyethylenimine (PEI; linear, 25 kD; Polysciences, IL, USA). To transfect 600 mL of HEK293F cells, 600 µg of plasmid and 1.2 mg of PEI were diluted into 30 mL of PBS (Gibco, Thermo Fisher Scientific, MA, USA) separately at room temperature (RT) for five minutes., followed by gentle mixing of the two solutions at RT for 20 min before transfection. The DNA-PEI mixture was introduced to the cell culture at a density of $1.3 \times 10^6$ cells·mL⁻¹. The transfected cells were cultured at 37 °C, 125 rpm, and in a humidified atmosphere with 8% $CO_2$ for 72 h. 100 mL of fresh FreeStyle™ 293 Expression Medium (Gibco, Thermo Fisher Scientific, MA, USA) was added to the culture to boost the recombinant protein expression additional three days prior to harvest.

The cell culture supernatants from both HEK293F and IPEC-J2 were collected by using centrifugation at 4540 × g and 4 °C for 30 minutes. Filtrate the supernatant by using a 0.2 µm filter (Thermo Fisher Scientific, MA, USA). Add 80 mL of 10 X binding buffer (200 mM Tris-HCl (pH 8.0), 1.5 M NaCl, 50 mM imidazole, 0.2% (w/v) $NaN_3$) to the

supernatant prior to binding with cobalt resin (Thermo Fisher Scientific, MA, USA) for overnight. The resin was washed by using wash buffer (20 mM Tris-HCl (pH 8.0), 300 mM NaCl, 10 mM imidazole, 0.02% (w/v) $NaN_3$) followed by elution with elution buffer (20 mM Tris-HCl (pH 8.0), 300 mM NaCl, 150 mM imidazole, 0.02% (w/v) $NaN_3$). The eluates were subsequently concentrated by a 100 kDa MWCO concentrator (Merck, MO, USA) and then applied to a size-exclusion column (Superose® 6 Increase 10/300 GL, Cytiva, MA, USA) in SEC buffer (50 mM Tris-HCl (pH 7.4), 150 mM NaCl, 0.02% (w/v) $NaN_3$). All fractions containing spike protein were collected and concentrated. The concentration of the spike was determined based on the absorbance at 280 nm using NanoPhotometer® N60 (IMPLEN, Germany).

### Negative stain EM for sample screening
Eight microliters of purified viral particles were applied to the carbon film 300 mesh copper grids (Electron Microscope Sciences, PA, USA), which were glow discharged with the PELCO easiGlow Glow Discharge Cleaning System (Ted Pella, CA, USA). The grids were subsequently stained with 8 µL of 2% uranyl formate (UF; Polysciences, PA, USA) for 1 min. The viral particles on the grids were observed under the FEI Tecnai G2 Spirit Transmission Electron Microscope (Field Electron and Ion Company, OR, USA) for screening.

### Cryo-ET sample preparation
80 µL of 10 nm gold fiducial (Merck, Darmstadt, Germany) was pre-incubated with 20 µL of 0.2 mg/mL BSA (Bionovas, Ontario, Canada) for 1 hr on ice then concentrated at 15,870 × g, 4 °C for 10 min. 80 µL of supernatant was carefully removed without disturbing the pellet, which was then resuspended to get the concentrated gold fiducial solution stock. The purified PEDV sample was mixed with concentrated gold fiducial solution in 1:1 ratio. 4 µL of the mixture was applied on Quantifoil R2/2 300 mesh holy carbon copper grids (Ted Pella, CA, USA) which were glow discharged at 25 mA for 30 s. The grids were blotted by using Vitrobot Mark IV (Thermo Fisher Scientific, MA, USA) with a blot force of 0, 3 s of blotting time at 4 °C and 100% humidity. The grids were then plunge frozen into liquid ethane and stored in liquid nitrogen until imaging.

### Cryo-ET tilt series acquisition and tomogram reconstruction
The sample grids were loaded into a 300 keV Titan Krios TEM instrument (Thermo Fisher Scientific) equipped with a K3 direct electron detector (Gatan, CA, USA). The cryo-ET tilt series were collected by Tomography 4.0 software (Thermo Fisher Scientific) in a bi-directional tilt scheme: sample was tilted from 20 to −61 degrees, then from 23 to 59 degrees, with nine frames acquired at every 3-degree step. Among the 90 tilt series, 56 had 100 $e^-/Å^2$ accumulative doses and 1.5 to 4 µm defocus, and the rest were acquired with 5 to 6 µm defocus and 120 $e^-/Å^2$ accumulative doses. All frames were motion-corrected and dose-filtered with IMOD 4.11.6 alignframes function and combined to each tilt series via IMOD newstack function[68,69]. Tomogram reconstruction was performed with IMOD - ETOMO[68]. The tomograms were binned four times (bin4, 5.558 Å/pixel) for the subsequent processing.

### Subtomograms averaging
To increase the contrasts of the S protein images, a 5 nm lowpass filter was applied to tomograms with EMAN2 e2proc3d utility[70]. For the initial template generation, tomograms were visualized under Chimera 1.13 for manual picking of S proteins[71]. A pair of coordinates were defined for each identified S protein using the Chimera Volume Tracer plugin: one to mark the top and the other to mark the bottom of the S-trimer, where the S-trimer connects to the viral membrane. An in-house MATLAB (ver. R2020a, MathWorks, MA, USA) script (initial_template_generation.m, available on Github) was used to calculate (i) the midpoints between the pair of coordinates, which we treated as the particle location, and (ii) the orientation of the vector from one to

the other coordinates in the pair, in Euler angles, as the particle orientation. The computed coordinates were rearranged to a table format readable by Dynamo 1.1.514 software[72], which we used for most STA-related tasks, if not otherwise stated. All S-trimer subtomograms were extracted with 64 pixels (356 Å) box size. Subtomograms were extracted from the tomograms according to the calculated coordinates. A low-resolution initial template of the S protein was generated after a few rounds of STA.

Details of the particle search procedure were described in the result section. The centers and radii of the viruses were measured with the vesicle model functions in Dynamo. 3D position and the orientation of subtomograms were visualized on tomograms with the PlaceObject plugin on Chimera[73]. Since the initial subtomogram coordinates were oversampled, over STA iterations subtomograms will cluster towards the real S protein locations. To remove subtomograms containing the same S proteins, a distance threshold was applied to remove subtomograms such that only the one with the highest cross-correlation to the given template remained in each subtomogram cluster. In addition, we observed that subtomograms tend to align towards the high-contrast viral membrane and ice-carbon boundaries. To resolve the membrane alignment, we calculated the dot product between the orientation vector of the membrane coordinates and the vector originating from that membrane coordinate to the closest S protein coordinates and any S protein coordinates with negative or close to zero dot products, i.e., the S proteins that are enclosed in the viruses or in the membrane, were removed. To remove the S proteins that align to the ice-carbon boundaries, we defined a boundary model in 3D to exclude the S protein coordinates that are near the boundary or on carbon. S protein subtomograms that had unrealistic tilt angles with respect to the closest membrane coordinates were removed. The tilt angle of the S protein to the normal of the membrane was calculated with another custom-built MATLAB script (spike_tilt_angle.m, available on Github), and the S protein subtomograms with more than 130° tilt angle were removed.

15,065 S protein subtomograms remained after cleaning. Particle subboxing was performed such that three smaller S protomer subtomograms with 32 pixels (178 Å) box size were generated from each S trimer subtomograms. PCA and MRA were performed as described in the result section to classify the S protomer subtomograms according to their D0 conformations. All protomers were mapped back to their original trimers, and all trimers were classified according to their number of D0-up protomers. S trimers that contained protomers that were classified as D0-mix or noise were discarded. 2385, 2747, 1355, 211 S trimer subtomograms were classified to $D0_{DDD}$, $D0_{UDD}$, $D0_{UUD}$, and $D0_{UUU}$ subsets, respectively. Subtomograms in each subset were reoriented such that their relative orientation of D0-up/down matches. For example, among the set of subtomograms that were classified as $D0_{UDD}$, trimers in the subtomograms might be arranged with UDD, DUD or DDU permutations. Reorientation was applied to the subtomograms so that all of them have their D0-up pointing in the same direction. Each subset was further refined, as described in the result section, to generate the final structure.

The alignment masks were created with the RELION mask create[74] at the later STA iterations. B-factor sharpening and resolution determination were performed using RELION post-processing. A homology model of the postfusion PEDV S trimer was built by SWISS-MODEL[75], using the postfusion S trimer structure of murine hepatitis virus (MHV; PDB: 6B3O)[45] as the template.

### Cryo-EM sample preparation and data collection
Four microliters of the purified PEDV sample were applied on glow-discharged (25 mA for 15 s) Quantifoil R1.2/1.3 300 mesh holy carbon copper grids (Ted Pella, CA, USA). The grids were blotted and plunge frozen in the same way as for cryo-ET. Grids were imaged using the same microscope as used for cryo-ET, operating in counting mode at a

nominal magnification of 64,000x, corresponding to a super-resolution pixel size of 0.7 Å/pixel. In total, 3701 movies were collected by EPU 2.0 (Thermo Fisher Scientific, MA, USA) with defocus ranging between 1 and 3 μm and accumulated exposure of 55.4 e⁻/Å² distributed over 50 frames.

For the cryo-EM analysis of the recombinant isolated PEDV PT52 S four microliters of 1.5 mg/mL, 1.3 mg/mL, or 1.1 mg/mL of the IPEC-J2 derived protein, HEK293F derived WT, or T236I mutant, respectively, were vitrified on glow-discharged (25 mA for 15 s) Quantifoil R1.2/1.3 or R2/1 300 mesh holy carbon copper grids (Ted Pella, CA, USA). The images were collected using the same microscope as the previous datasets, except a magnification of 81,000x was used, corresponding to a super-resolution pixel size of 0.53 Å/pixel. Data were recorded at 0° and 30° stage tilt in order to compensate for the preferred orientation adopted by the PEDV PT52 S particles in the ice. In total, for the IPEC-J2 derived protein 2419 and 3387 movies; for the HEK293F derived WT protein we collected 946 and 1628 movies; and for the HEK293F derived T326I mutant 640 and 1787 of the un-tilted and tilted movies, respectively. In all three datasets, the set defocus ranged between 1.8 and 2.3 μm, and accumulated exposure was around 50 e⁻/Å² distributed over 50 frames.

### Cryo-EM image processing and model building

Micrograph movies of the PEDV S on intact virus particles were aligned using MotionCor2 (UCSF)[76] and CTF corrections were achieved by CTFFIND4.1[77]. The particle images of the S proteins on the virus particles were manually picked and used as a training set for automated particle picking by Topaz convolutional neural network[78]. Further data processing was performed in RELION-3.1[74], while the final non-uniform refinement and variability analysis were performed in CryoSparc 3.1 (Structura Biotech, Ontario, Canada)[49,79]. After extracting particle images, the dataset was cleaned by 3D classification using low-pass-filtered map of the reported structure of PEDV S trimer[31], as a starting reference. Selected 75,810 best particles were refined with a C3 symmetry imposed and symmetry expanded[80]. This method in combination with a mask focused on a single PEDV S protomer and 3D variability analysis algorithm in CryoSparc 3.1[81] allowed us to computationally isolate individual S protomers per particle image and evaluate their conformational heterogeneity. The analysis revealed existence of two alternative states (D0-up and D0-down) of the PEDV S protomer. The best aligning protomer particles representing the two conformations were selected, of which 40,542 and 33,759 particles corresponded to D0-down and D0-up conformations, respectively. Protomer particles were mapped back to the trimer particles, and the datasets were cleaned such that any duplicate PEDV S particle images that could be introduced by the symmetry expansion procedure were removed before following refinements. The particle images were then submitted for another round of 3D classification to identify the highest quality S protein images in each dataset. The final images stacks contained 19,350 and 9319 particles in the D0-down and D0-up datasets, respectively. The trimer particles that only contained D0-down protomers were 3D-refined with C3 symmetry, yielding a cryo-EM map of $D0_{DDD}$ PEDV S trimer with 4.7 Å global resolution. The trimer particles containing at least one D0-up protomers were 3D-refined with no symmetry constraint, resulting in a cryo-EM map of $D0_{UDD}$ PEDV S trimer with 6.4 Å global resolution. Further 3D classification and 3D variability analysis of S trimers with D0-up protomers produced $D0_{UUD}$ and $D0_{UUU}$ PEDV S trimer cryo-EM maps. The details of the data processing are shown in Supplementary Fig. 3.

The cryo-EM structural analysis of the recombinant PEDV PT52 S expressed in IPEC-J2 cell line was performed entirely in CryoSparc 3.1[49,79] except motion correction, which was done using MotionCor2 (UCSF)[76]. The micrographs were then CTF corrected using Patch CTF within CryoSparc 3.1. For the initial steps of the processing, the particle images were extracted with 2-fold down sampling, while

the final refinements were done with full-resolution data. The dataset was cleaned by 2D classification, leaving 396,287 good particle images. Next, heterogeneity of the trimer composition was resolved by 3D classification/heterogenous refinement, which resulted in a map of the PEDV PT52 S trimer in $D0_{UUD}$ conformation at a global resolution of 4.2 Å refined with 50,949 particle images and $D0_{UUU}$ conformation at a global resolution of 3.1 Å refined with 298,195 particle images. To better resolve densities representing the dynamic D0 and CTD domains of the PEDV S, both of the S trimer conformations were refined with C3 symmetry imposed and then symmetry expanded[80]. The 3D classification (without alignment) focused on an individual protomer and guided by volumes generated by 3D variability analysis revealed D0-up protomer conformations with the CTD in closed/down and open/up position producing maps at a resolution of 3.2 Å and 3.3 Å, respectively. The details of the data processing pipeline are shown in Supplementary Fig. 8.

The micrograph movies of the recombinant isolated PEDV PT52 S derived from HEK293F cells, both WT and T326I mutant, was motion and CTF corrected in a similar way as the movies of intact virus particles. The particles were automatically picked based on 2D class averages of manually picked PEDV S WT particles using RELION-3.1[74]. Particle images were extracted within the same software, while the following 2D and 3D classification and refinements were performed in CryoSparc 3.1[49,79]. After dataset cleaning by 2D classification, the WT protein dataset consisted of 95,371 particles, while the T326I dataset consisted of 99,985 particles. After resolving heterogeneity within the dataset using 3D classification/heterogenous refinement, we obtained maps of PEDV PT52 S WT in $D0_{DDD}$ conformation at a global resolution of 5.6 Å refined with 10,676 particle images; $D0_{UDD}$ conformation at a resolution of 10.2 Å refined with 6,446 particle images; and $D0_{UUU}$ conformation at a resolution of 4.5 Å refined with 78,249 particle images. Following similar data processing pipeline, in the PEDV PT52 S T326I dataset, we identified $D0_{DDD}$; $D0_{UDD}$; $D0_{UUD}$; and $D0_{UUU}$ conformations producing maps at a resolution of 5.2 Å; 8.0 Å; 6.1 Å; and 6.2 Å, and refined with 42,239; 15,682; 19,273; and 22,791 particles images, respectively. More detailed data processing steps are shown in Supplementary Fig. 11 and Supplementary Fig. 12.

Atomic models of all of the PEDV PT52 S protein conformations reported herein were built in Coot[82] and ChimeraX[83] plugin, ISOLDE[84] and, while final refinement was performed in PHENIX[85] based on the homology model obtained by SWISS-MODEL[75], which was used previously determined structure of the S trimer[12,31] as a template. Glycan structures were built manually Coot[82] and refined in PHENIX[85] based on the densities observed in the cryo-EM maps.

### Glycopeptide analysis

Purified recombinant PEDV S proteins were further separated by running 10% SDS-PAGE for 15 min at 80 V followed by 40 min at 180 V, then stained by Coomassie Blue. The target protein band corresponding to the PEDV S protein was taken for in-gel reduction by 10 mM dithiothreitol (DTT) at 37 °C for one h, alkylation by 50 mM iodoacetamide for one h in the dark at RT, and then digestion by chymotrypsin (Promega, Madison, WI) overnight at 37 °C. All reactions were performed in 25 mM ammonium bicarbonate (ABC) buffer. The glycopeptides were extracted and cleaned up by ZipTip C18 (Merck Millipore, MO, USA) for identification.

Analysis of glycopeptide was performed on an EASY-nLC™ 1200 system fitted to a Thermo Orbitrap Fusion Lumos mass spectrometer (Thermo Fisher Scientific, Bremen, Germany). The glycopeptide mixture redissolved in 0.1% formic acid (Solvent A) was loaded onto a 25 cm PepMap C18 column (Thermo Fisher Scientific, MA, USA) and separated using a gradient of 5–45% solvent B (80% acetonitrile with 0.1% formic acid) in 75 min at a flow rate of 300 nl/min. The MS1 survey scan spectra were acquired from 400 to 1800 m/z in the Orbitrap at 120 K resolution with an AGC target of $2 \times 10^5$ ion count.

The top 15 most intense precursors were isolated in the quadrupole with a 2 Da isolation window for HCD product-dependent EThcD MS2 fragmentation. HCD MS$^2$ were acquired at 28% normalized collision energy in the Orbitrap at 30 K resolution, with the AGC target set to $5 \times 10^4$ and the maximum injection time 65 ms. EThcD MS2 was performed by calibrated charge-dependent parameters supplemented with HCD at 15% collision energy, which was triggered by the detection of one of the three HCD product ions at m/z 138.0545, 204.0867, 366.1396. The AGC target for EThcD-MS$^2$ was set at $3 \times 10^4$, acquired in the Orbitrap at a 60 K resolution.

The raw data were processed by Byonic (v3.9.6 for HC070225, and v4.3.4 or v4.0.12 for PT52), Byos with Byologic (v4.0.53 for HC070225 and v4.4.74 for PT52) (Protein Metrics Inc.), and pGlyco 3.0[86]. The MS$^2$ data were searched against the PEDV S protein sequence and the built-in N-glycan database (182 human no multiple fucose for spike protein derived from HEK293 cells and an additional 309 mammalian no sodium for those produced by IPEC J2 cells) for Byonic and the default N-glycan database (pGlyco-N-Human for HEK293-derived proteins and pGlyco-N-Mouse-large for IPEC J2-produced proteins) for pGlyco3.0[86]. Fully specific cleavages at R, K, F, Y, W, and L residues with up to two missed cleavages were allowed for enzyme digestion. The tolerance for precursor and fragment ions was set at 5 and 10 ppm, respectively. Cysteine carbamidomethylation was set as a fixed modification, whereas oxidation and deamidation were set as variable modifications. Further data mining was performed by an in-house script, with outputs from Byonic, Byos, and pGlyco3 collated, aligned, and grouped accordingly using the following filtering criteria: Score > 200, PEP2D < 0.001 for Byonic, and PepScore > 5, GlyScore > 4 for pGlyco3. The same glycopeptides identified by both Byonic and pGlyco3 were accepted as high-confidence hits, whereas those identified by one but not the other were further manually verified. Identified glycopeptides were quantified based on the area under the curve of their extracted ion chromatogram (XIC-AUC) by Byologic and pGlyco3 to determine the most abundant glycoforms for each site, as presented in Supplementary Table 1. In cases where differences were found between duplicate samples derived from HEK293 or IPEC J2 cells, the larger (more complex) glycan structure was used to represent that site in the model building.

### Reporting summary

Further information on research design is available in the Nature Research Reporting Summary linked to this article.

## Data availability

The Atomic coordinates of PEDV PT52 S determined in situ on the viral particles generated in this study have been deposited in the Protein Data Bank (PDB) under the accession codes 7W6M (prefusion D0$_{DDD}$ structure), 7W73 (prefusion D0$_{UDD}$ structure); The Atomic coordinates of the IPEC-J2 cell-produced recombinant PEDV PT52 S generated in this study have been deposited in the Protein Data Bank (PDB) under the accession codes 7Y6S (prefusion D0$_{UUU}$ structure), 7Y6T (prefusion D0$_{UUD}$ structure), 7Y6U (single protomer of D0-up with CTD-close) and 7Y6V (single protomer of D0-up with CTD-open). The cryo-EM maps generated in this study have been deposited in the Electron Microscopy Data Bank (EMDB) under the accession codes EMD-32329 (prefusion in situ D0$_{DDD}$ structure, EMD-32338 (prefusion in situ D0$_{UDD}$ structure), EMD-33646 (prefusion IPEC-J2 D0$_{UUU}$ structure), EMD-33647 (prefusion IPEC-J2 D0$_{UUD}$ structure), EMD-33648 (single protomer of D0-up with CTD-close), and EMD-33649 (single protomer of D0-up with CTD-open), EMD-33702 (prefusion HEK293F D0$_{UUU}$ structure), EMD-33701 (prefusion HEK293F D0$_{UDD}$ structure), EMD-33700 (prefusion HEK293F D0$_{DDD}$ structure), EMD-33706 (T326I prefusion HEK293F D0$_{UUU}$ structure), EMD-33705 (T326I prefusion HEK293F D0$_{UUD}$ structure), EMD-33704 (T326I prefusion HEK293F

D0$_{UDD}$ structure), and EMD-33703 (T326I prefusion HEK293F D0$_{DDD}$ structure). The cryo-ET maps generated in this study have been deposited in EMDB under accession codes EMD-32332 (prefusion in situ D0$_{DDD}$ structure), EMD-32333 (prefusion in situ D0$_{UDD}$ structure), EMD-32337 (prefusion in situ D0$_{UUD}$ structure), EMD-32339 (prefusion in situ D0$_{UUU}$ structure), EMD-32340 (postfusion in situ structure). The LC-MS/MS raw datasets of IPEC-J2 cells-derived PEDV PT52 S, HEK293 cells-derived PEDV PT52 S, and JEK293 cells-derived PEDV HC070225 S generated in this study have been deposited to MassIVE under accession code MSV000088544. The same mass spectrometry proteomics data have been deposited to the ProteomeXchange Consortium via the PRIDE[87] partner repository with the dataset identifier PXD035778 and https://doi.org/10.6019/PXD035778 [https://www.ebi.ac.uk/pride/archive/projects/PXD035778].

## Code availability

The code used in this study is public available in Github [https://doi.org/10.5281/zenodo.6945928].

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

## Acknowledgements

This work was supported by Academia Sinica intramural fund, an Academia Sinica Career Development Award, Academia Sinica to S.T.D.H. (AS-CDA-109-L08), the Infectious Disease Research Supporting Grants to S.T.D.H. (AS-IDR-110-08 and AS-IDR-111-03), and the Ministry of Science and Technology (MOST), Taiwan (MOST 109-3114-Y-001-001, MOST 110-2113-M-001-050-MY3 and MOST 110-2311-B-001-013-MY3) to S.T.D.H., (MOST 110-2811-B-001-560-) to P.D., and (MOST109-2313-B-002-016-MY3) to H.W.C. We thank the Academia Sinica Biophysics Core Facility (AS-CFII108-111), Academia Sinica Common Mass Spectrometry Facilities (AS-CFII-108-107), and Academia Sinica Cryo-EM Center (AS-CFII-108-110 and AS-CFII-111-210) for data collection, all of which are funded by the Academia Sinica Core Facility and Innovative Instrument Project. Taiwan Protein Project (AS-KPQ-109-TPP2) is also acknowledged for supporting the cryo-EM facility. We also thank the mammalian cell culture facility of Institute of Biological Chemistry, Academia Sinica, for supporting the protein production.

## Author contributions

S.T.D.H.—conceptualization; C.Y.C., T.J.Y., Y.S.W., Y.H.C., Yen-Chen Chang—sample preparation; C.Y.H., P.D., Y.M.W., C.H.W., Yuan-Chih Chang, Yu-Chun Chien, Y.S.W.—data collection; C.Y.H., P.D., Yu-Chun Chien, Y.S.W., Y.X.T.—data analysis; H.W.C., K.H.K., S.T.D.H.—funding acquisition; C.Y.H., P.D., K.H.K., H.W.C., S.T.D.H.—methodology; C.Y.H., P.D., C.Y.C., Y.S.W., S.T.D.H.—manuscript writing, review, and editing.

## Competing interests

The authors declare no competing interests.
