## [Peer Review File · Nature Communications]

In situ structure and dynamics of an alphacoronavirus spike protein by cryo-ET and cryo-EMReviewers' Comments:

Reviewer #1:

Remarks to the Author:

PEDV is an alphacoronavirus that causes digestive disorders in pigs, and infections in neonatal piglets can result in high case-fatality rates. The spike protein is responsible for virus attachment and membrane fusion, and high-resolution single-particle cryo-EM structures of PEDV spikes from two groups resulted in two different conformations of the spike, wherein domain 0 was either in an all-down or all-up conformation. Here, Huang et al investigate the structural arrangement of domain 0 within spikes on viral particles by cryo-ET and cryo-EM. The results from their cryo-ET studies revealed that 35.6% of the S proteins adopted an all D0-down arrangement, 41.0% of the S proteins adopted a one D0-up and two D0-down arrangement, 20.2% of the S-proteins exhibited a two D0-up and one D0-down arrangement, and 3.2% of the S proteins exhibited an all D0-up arrangement. This is the first time that mixed populations of PEDV spikes have been reported. Interestingly, the distribution of the spike conformations observed agrees very well with the distribution expected if each protomer had a probability of being in the up conformation 30% of the time, which is the value obtained from the cryo-ET analysis of individual protomers. This indicates that the up-down conformational change of one protomer is independent of the others. The authors also perform cryo-EM studies on the viral particles to obtain higher-resolution structures of the PEDV spikes, resulting in both a 3-down and a 1-up-2-down reconstruction. A mass-spec-based analysis of the N-linked glycans was also performed to aid model building.

This is a strong manuscript that thoroughly investigates the conformation of PEDV spikes on viral particles. The EM studies are performed well, and the results provide new insights into the domain 0 conformational dynamics. The authors do not provide a molecular basis for the conformational changes, however, this may be beyond the scope of the manuscript. Future studies could investigate changes in spike dynamics resulting from site-directed mutagenesis experiments. The figures are excellent and the writing is generally clear, although there are some grammatical mistakes throughout the text that should be addressed. A few are noted below.

Minor Edits:

1. Intro, 1st paragraph: entropathogenicity should be enteropathogenicity
2. Results: "(v) removing particles that has unrealistic tilt angle" ; should be 'have'
3. Results: "without imposing any symmetry constrain" ; should be 'constraints'
4. Figure 3F is called out in the results section, but it appears that it should have been 3G: "and k underwent a 93° rotation away from the S2 subunit pivoting at the hinge between the D0 and NTD (Figure 3F)."

Reviewer #2:

Remarks to the Author:

The manuscript by Cheng-Yu Huang et al, entitled „In situ structure and dynamics of an alphacoronavirus spike protein by cryo-ET and cryo-EM“ uses subtomogram averaging and single-particle reconstruction to investigate the conformational landscape of PEDV S glycoprotein. The manuscript is well written, and the data are well presented, however, the results section contains too many details about image processing and overall, the study brings only a few results. While it is important to structurally characterize the S spike of PEDV in situ, the manuscript does not deliver a sufficient number of novel findings in coronavirus biology. There are already high-resolution studies of PEDV S spike showing that D0 might be either 'Up' or 'Down' in different PEDV strains (Kirchdoerfer et al, 2021; Wrapp D et al, 2019) and it is well known that coronavirus S spike can tilt. Although the manuscript adds information on the S D0 flexibility of alphacoronaviruses it is rather descriptive.

Major concerns:

Since the D0 modulates the enteric tropism of PEDV by binding to sialic acids on the surface of enterocytes one would expect that D0 should be in "Up" conformation in the presence of sialic acid moieties, however, these experiments have not been performed. D0 movement could be an artefact of preparation and since the study does not provide any evidence that S D0 movement is required during virus entry or immune escape.

The manuscript provides only descriptive structures of S spike conformational changes of the D0 domain that is found in alphacoronaviruses. As the authors mention the fraction of up or down conformations may depend on sample preparation, chemical fixation used for virus inactivation or even image processing method as shown in the discrepancy between the STA and SPR. Therefore, it is difficult to deduce any conclusions and only speculations regarding the physiological role of up and down conformation are provided. In addition, a direct comparison of DDD, UUD, UUU and UUU conformations obtained by cryo-ET and cryo-EM is missing. In the cryo-EM section (page 7), only 68% of DDD is presented. A plot such as the one presented in 1F for the cryo-EM dataset would be beneficial to present the data.

Authors propose that N-glycosylation plays a role in stabilizing the D0-up conformation. While this is interesting and could be tested by introducing a mutation into the S spike and performing a SPA, such an experiment was not done. The structure of S with neutralizing antibodies are not shown and it is not clear how this data will help in designing a more effective vaccine against PEDV.

In addition, no information is provided if any biosafety measures were observed during the propagation of the virus which is highly pathogenic in pigs.

Reviewer #3:

Remarks to the Author:

In the manuscript submitted by Huang et al. the authors describe the application of cryo-ET and cryo-EM to investigate the structure and structural dynamics of a porcine epidemic diarrhea virus (PEDV) spike glycoprotein from a strain which is particularly lethal in neonatal piglets from either naïve or vaccinated herds, Pintung 52. This group of coronaviruses contains an additional domain not seen in other coronaviruses such as SARS-CoV-2, the D0 domain. The authors investigate the dynamics of this domain and demonstrate that, unlike previous studies which select for particular conformations, there are multiple conformations of D0 possible on a single spike glycoprotein. In addition, the authors propose a model for the transition of the D0 domain from 'up' to 'down'. Finally the authors investigate the N-linked glycans found on PEDV, using a separate strain of PEDV as the one investigated did not give high yields of protein. This analysis revealed a range of different glycan processing states, as can be seen on a range of coronavirus spike proteins. Overall the study is an interesting study investigating the structure of infectious virions, and the observations are of importance to vaccine design for PEDV, from the perspective of D0 enabling viral immune escape. My expertise is in the analysis of the glycosylation of viral glycoproteins, and as such I will focus most of my comments on this aspect of the manuscript, although, to my non-expert reading, the EM analysis appears well considered and follows a logical progression. My main comment on the EM data is that the authors correctly emphasize the presence of post-fusion spikes on infectious virions, and hypothesize that "The formation of the postfusion state could be attributed to the addition of trypsin in the cellular culture medium during the viral particle production". This raised a question about cleaved versus uncleaved spike protein on the mature virion. In HIV-1 Env and SARS-CoV-2, furin cleavage is required to produce infectious virions. Certain cell lines can contain low endogeneous furin and as a result, uncleaved material can be present on the surface of the virion. In HIV-1 Env, this uncleaved material can have distinct conformations compared to properly cleaved and folded Env. I was wondering if the authors had considered the potential for uncleaved misfolded S protein to be present in their datasets and if they controlled for this? One potential way would be to dissociate the non-covalent interactions between S1 and S2 and investigate the proportion of spike proteins that remains intact, as this would suggest a lack of cleavage in the peptide backbone. There is potential for uncleaved material to be impacting the observed abundances of D0. If the authors could acknowledge this potential and, if feasible, perform a few additional experiments then that would be really

interesting to know.

From a glycobiology perspective, this reviewer certainly acknowledges the difficulties in obtaining sufficient material, and the use of a distinct spike glycoprotein is understandable. However, I feel that there are certain aspects to the glycan analysis that could be improved to provide a deeper insight into the glycosylation of PEDV. The authors describe the predominant glycan structure detected at each site in Figure 4. The materials and methods describe the process of quantification as "Identified glycopeptides were quantified based on the area under the curve of their extracted ion chromatogram (XIC-AUC)". As such there is likely more information that could be presented regarding the distribution of N-linked glycans on the S protein. Other coronavirus glycoprotein analyses demonstrate the different N-linked glycan compositions using XIC AUC analysis (<https://doi.org/10.1016/j.eng.2020.07.014> , <https://doi.org/10.1016/j.chom.2020.08.004>, doi: 10.1126/science.abb9983). This study would benefit from including the distributions of different N-linked glycans as N-linked glycan processing is an extremely heterogeneous process, and presenting more of the glycan data would provide more information about the structure-function relationship between the D0 and the glycans.

Additionally, the authors should acknowledge the differences in glycosylation resulting from different cell lines. Many conditions contribute to the diverse range of N-linked glycans on mature human glycoproteins, and the production of virus/recombinant protein either in Vero cells (monkey origin) or HEK cells (human origin) will be different to that observed in pigs. For example, in addition to NeuAc, other mammals modify their N-linked glycans with NeuGc, which is not present to the same extent in HEK or Vero cells. Additionally Vero and HEK cells process glycans differently to one another (<https://doi.org/10.1021/acs.biochem.1c00279>) and it has been shown that Vero cell-derived glycans display relatively low levels of sialylation compared to other cells (DOI: 10.4149/av_2011_02_147). The authors should acknowledge these differences in the manuscript. If feasible the ideal outcome would be the production of virus in a porcine cell line and subsequent glycan analysis, although this may likely be beyond the scope of the project.

Another point regarding the glycan analysis is the choice of sample to analyse. Whilst the selected strain has a high sequence homology to the strain studied by EM, there is an extra glycan site at position N723. As the authors state in the discussion: "In contrast, the D0-down conformation exhibited a cluster of N-glycans on Asn726, Asn873, and Asn118, which made extensive contacts with the D0 and S2 subunit". The presence of an N-linked glycan site so close to another will likely have impacts on the processing of the other, as can be seen on HIV-1 Env (<https://doi.org/10.1016/j.isci.2020.101711>). As a glycan site proximal to the D0 domain, I would encourage the authors to delete the N723 glycan site from the strain analysed, and investigate how the glycan processing compares. I feel that this would be a more appropriate model to use for the structures presented in this paper.

My final point is regarding the difficulty expressing and purifying coronavirus S protein from the strain analysed in the manuscript. Recent work building on previous coronavirus studies from the McLellan lab, including a 3.1 angstrom structure of PEDV, have produced high yields of spike protein using proline substitutions. Whilst not applicable for the structural studies reported in this manuscript, this approach may enable protein to be produced corresponding to the Pintung 52 strain.

To summarise the above, I feel that the current manuscript is a good fit for a specialist structural biology journal, however much more information can be gained from a deeper understanding of the glycan shield and how it influences the D0 conformation. An ideal approach would be the production of a sequence matched protein in a porcine cell line. If this is not possible then employing the mutations outlined above would be a good second option. If not then I think it is important to investigate the impacts of the deletion of the N723 site on the glycosylation of the remaining sites. Despite the authors instructions, I was unable to access the raw data from the MASSive server using the reviewer instructions, this could be an error on my end, but I would encourage the authors to make the dataset public as soon as possible. The methods used for glycan analysis appear sound as written.

Minor comments:

The introduction may benefit from some background regarding glycosylation and how it is important for protein structure

In addition, some studies have reported potential receptors for PEDV, see ref 13 in the manuscript,

"PEDV has been reported to utilize the aminopeptidase N protein (APN, also known as CD13) as a functional receptor"

Please find below the point-by-point responses to the Reviewers' comments. All modifications in the manuscript are highlighted in red. We hope the Reviewers will find these revisions acceptable for publication in *Nature Communications*.

Reviewer #1 (Remarks to the Author):

PEDV is an alphacoronavirus that causes digestive disorders in pigs, and infections in neonatal piglets can result in high case-fatality rates. The spike protein is responsible for virus attachment and membrane fusion, and high-resolution single-particle cryo-EM structures of PEDV spikes from two groups resulted in two different conformations of the spike, wherein domain 0 was either in an all-down or all-up conformation. Here, Huang et al investigate the structural arrangement of domain 0 within spikes on viral particles by cryo-ET and cryo-EM. The results from their cryo-ET studies revealed that 35.6% of the S proteins adopted an all D0-down arrangement, 41.0% of the S proteins adopted a one D0-up and two D0-down arrangement, 20.2% of the S-proteins exhibited a two D0-up and one D0-down arrangement, and 3.2% of the S proteins exhibited an all D0-up arrangement. This is the first time that mixed populations of PEDV spikes have been reported. Interestingly, the distribution of the spike conformations observed agrees very well with the distribution expected if each protomer had a probability of being in the up conformation 30% of the time, which is the value obtained from the cryo-ET analysis of individual protomers. This indicates that the up-down conformational change of one protomer is independent of the others. The authors also perform cryo-EM studies on the viral particles to obtain higher-resolution structures of the PEDV spikes, resulting in both a 3-down and a 1-up-2-down reconstruction. A mass-spec-based analysis of the N-linked glycans was also performed to aid model building.

This is a strong manuscript that thoroughly investigates the conformation of PEDV spikes on viral particles. The EM studies are performed well, and the results provide new insights into the domain 0 conformational dynamics. The authors do not provide a molecular basis for the conformational changes, however, this may be beyond the scope of the manuscript. Future studies could investigate changes in spike dynamics resulting from site-directed mutagenesis experiments. The figures are excellent and the writing is

generally clear, although there are some grammatical mistakes throughout the text that should be addressed. A few are noted below.

The authors do not provide a molecular basis for the conformational changes, however, this may be beyond the scope of the manuscript. Future studies could investigate changes in spike dynamics resulting from site-directed mutagenesis experiments.

We thank the Reviewer for the positive review and summary of our work. As suggested by the Reviewer and the other two Reviewers, we have carried out site-directed mutagenesis with the human HEK293 Freestyle cell line to generate the recombinant PEDV PT52 S protein with the matching sequence as used in the intact virus, and the variant with the T326I mutation, to investigate the molecular mechanism underlying the regulation of the D0 motion as observed in the intact virus. The N-glycan at residue N324 was partially visualized *in situ* by cryo-EM, and it was located at the hinge region that connects the D0. We speculated in our original manuscript that this N-glycan could help maintain the D0 in an up conformation. We chose to introduce the T326I mutation to remove the N-glycan at residue N324 by nullifying the NxS/T sequon required for N-glycosylation. This sequential difference is present between the original PEDV G1 strain CV777 and the more virulent CO/13 G2 strain as well as our PEDV PT52 strain (Figure S10E); therefore, it represents the natural difference between the different PEDV strains. Comparison of the cryo-EM maps of the recombinant PEDV PT52 S with and without the T326I mutation clearly showed that the loss of the N-glycan on Asn324 resulted in the increased D0-down population (Figures S14), thus confirming our hypothesis for the regulatory role of this N-glycan. Note that the recombinant PEDV PT52 S exhibited more D0_{UUU} arrangement as opposed to the dominant D0_{DDD} arrangement observed *in situ* in the intact virus. Nevertheless, we observed the same for the recombinant PEDV PT52 S produced in the porcine IPEC-J2 cell line, which closely resembles the native host cells for PEDV (Figure 5 and Figure S14). We should emphasize that this is the first time that a porcine cell line is used to generate recombinant proteins for structural and mass spectrometry-based glycosylation analyses. Our findings also highlighted the potential artifacts of using recombinant S proteins without the membrane attachment, as has been

widely used in many structural studies of coronavirus S proteins, including the ones for the two PEDV strains (<https://doi.org/10.1128/JVI.02078-14>, <https://doi.org/10.1073/pnas.1908898117>).

Intro, 1st paragraph: entropathogenicity should be enteropathogenicity

We thank the Reviewer for the careful proofreading. The typo on page 3, line 12 has been corrected.

Results: “(v) removing particles that has unrealistic tilt angle” ; should be ‘have’

The typo on page 5, line 27 has been corrected.

Results: “without imposing any symmetry constrain” ; should be ‘constraints’

The typo on page 6, line 10 has been corrected.

Figure 3F is called out in the results section, but it appears that it should have been 3G: “and k underwent a 93° rotation away from the S2 subunit pivoting at the hinge between the D0 and NTD (Figure 3F).”

The error on page 8, line 19 has been corrected.

Once again, I would like to thank the Reviewer for the careful proofreading. The revised manuscript has been proofread to identify and correct grammatical mistakes.

Reviewer #2 (Remarks to the Author):

The manuscript by Cheng-Yu Huang et al, entitled „In situ structure and dynamics of an alphacoronavirus spike protein by cryo-ET and cryo-EM“ uses subtomogram averaging and single-particle reconstruction to investigate the conformational landscape of PEDV S glycoprotein. The manuscript is well written, and the data are well presented, however, the results section contains too many details about image processing and overall, the study brings only a few results. While it is important to structurally characterize the S spike of PEDV in situ, the manuscript does not deliver a sufficient number of novel findings

in coronavirus biology. There are already high-resolution studies of PEDV S spike showing that D0 might be either 'Up' or 'Down' in different PEDV strains (Kirchdoerfer et al, 2021; Wrapp D et al, 2019) and it is well known that coronavirus S spike can tilt. Although the manuscript adds information on the S D0 flexibility of alphacoronaviruses it is rather descriptive.

Major concerns:

Since the D0 modulates the enteric tropism of PEDV by binding to sialic acids on the surface of enterocytes one would expect that D0 should be in “Up” conformation in the presence of sialic acid moieties, however, these experiments have not been performed. D0 movement could be an artefact of preparation and since the study does not provide any evidence that S D0 movement is required during virus entry or immune escape.

We thank the Reviewer for raising the issue of the functional relevance of the D0 orientation in the context of sialic acid-binding. Indeed, the sialic acid-binding activity of the D0 has been implicated in mediating host recognition by PEDV (<https://doi.org/10.1128/JVI.00430-15>, <https://doi.org/10.1016/j.virusres.2016.05.031>). However, there is no direct structural information to suggest that the D0-up conformation is obligatory for sialic acid-binding.

To address the Reviewer's comment, we sought to explore evidence for the D0-mediated sialic binding by generating two variants of the recombinant D0, which were isolated D0 and a D0 fused to the trimerization foldon sequence namely, D0F, used in the full ectodomain of the PEDV PT52 S to generate a stable trimer to yield increased multivalency. Both constructs have a His-tag at the C-terminus for affinity purification. Our attempt to produce the D0 variants using the HEK293 Freestyle cell line showed proper expression levels for both constructs followed by purification using immobilized metal affinity chromatography (IMAC). However, the subsequent size-exclusion chromatography (SEC) showed clear evidence of aggregation of both constructs (see figures below). The D0 domain contains several intra-domain disulfide bonds in addition to several free cysteines. Despite using the structural information derived from our cryo-

EM structure to guide the design of the isolated D0 constructs, the severe aggregation issue suggests the possibility that the D0 domain requires additional structural motives outside the D0 domain or auxiliary factors to facilitate its folding and maintain its solubility.

Purification results of both D0 and D0F

(A) Both D0 and D0F were purified by SEC using a Superdex 200 10/300 column. The UV curves of D0 and D0F are shown in black dash line and black solid line, respectively. The fractions No. 12-36 shaded in yellow were evaluated by SDS-PAGE (B). The protein bands corresponding to D0 and D0F are indicated by arrows.

The Reviewer’s remark that “...D0 movement could be an artefact of preparation and since the study does not provide any evidence that S D0 movement is required during virus entry or immune escape.” is duly noted. At the same time, it is technically very challenging to investigate the D0 movement during viral entry or immune escape *in situ*, which may require cryo-ET analyses of PEDV viral particles in the presence of host cells with and without sialic acids presenting on the host cell surface. Such a study is beyond the scope of the current work, and we shall carefully consider the possibility of conducting such experiments in the future. The focus of the current study, however, is to highlight

the intrinsic dynamics of the D0 as a part of the intact virus, which has not been documented before. Two previous studies report either all D0-up or all D0-down conformations for different PEDV strains, and we are the first to demonstrate the co-existence of the D0-up and D0-down conformations in a quantitative manner by cryo-ET and cryo-EM.

Indeed, the Reviewer is correct in stating that the D0-up conformation may be an artefact of preparation. Our cryo-EM analyses of the recombinant PEDV PT52 S expressed in the IPEC-J2, and HEK293 Freestyle cell lines showed identical structures in the all D0-up (D0_{UUU}) arrangement as the dominant population as opposed to the all D0-down (D0_{DDD}) arrangement observed *in situ*. Additionally, we observed the opening of the C-terminal domain (CTD) in the recombinant PEDV PT52 S, which was not observed *in situ* (Figure 5). The opening motion of the CTD highly resembles the receptor-binding domain (RBD) motions of SARS-CoV, SARS-CoV2, and MERS-CoV, which are well-documented. While these new findings underlined the potential pitfalls in studying the S protein structures using recombinant proteins, such experimental designs are necessary to facilitate high-resolution structural investigation, and the information derived from recombinant proteins have been instrumental in explaining the molecular mechanisms of a variety of coronaviruses, including SARS-CoV-2, and in developments of antiviral drugs against COVID-19. We, therefore, felt that the merits of our findings, especially the new additions of the recombinant protein analyses, should be considered.

The manuscript provides only descriptive structures of S spike conformational changes of the D0 domain that is found in alphacoronaviruses. As the authors mention the fraction of up or down conformations may depend on sample preparation, chemical fixation used for virus inactivation, or even image processing method as shown in the discrepancy between the STA and SPR. Therefore, it is difficult to deduce any conclusions and only speculations regarding the physiological role of up and down conformation are provided.

We agree with the Reviewer that this study did not provide direct evidence to link the D0 motion to the physiological role of this PEDV S. Nevertheless, we would like to reiterate that this study revealed the intrinsic propensity of the D0 to exist in a combination of up-

and down-conformations that has not been reported before. In fact, before the COVID-19 outbreak that highlighted the apparent need to understand the conformational diversity of the SARS-CoV-2 S protein to help design better validation experiments, few structural studies have gone to such great lengths to characterize the domain motions of other S proteins. On the one hand, the new cryo-EM results of the recombinant PEDV PT52 S variants with site-directed mutagenesis enabled us to make functional attribution of the N-glycan on Asn324 in regulating the D0 conformation. On the other hand, these findings indeed highlighted the intrinsic limitation and potential artefacts of structural studies using recombinant proteins. While the community should acknowledge such a pitfall, the use of recombinant proteins for detailed structural and functional characterizations remains essential for the literature. We hope the publication of our findings will attract the attention of virologists and veterinary scientists who are better equipped for physiological investigations to make further validations of the physiological role of the D0 and the CTD conformations.

In addition, a direct comparison of DDD, UUD, UUU and UUU conformations obtained by cryo-ET and cryo-EM is missing. In the cryo-EM section (page 7), only 68% of DDD is presented. A plot such as the one presented in 1F for the cryo-EM dataset would be beneficial to present the data.

As per the request from the Reviewer, we show below the correlation plot of the theoretical and experimental populations of different D0 arrangements derived from cryo-EM SPR analysis. The apparent deviation between the two can be explained by the priority of selecting subsets of particle images to generate high-resolution 3D maps. As such, many particle images were excluded from the datasets. Such a stringent filtering criterion was not applied to cryo-ET data, as we have noted on page 12, first paragraph, so that the relative populations of the different D0 arrangements derived from cryo-ET would be more representative and consistent with the theoretical values. This is indeed an issue associated with the data processing procedures, and we also address this issue in our manuscript on page 12, lines 1-6.

The correlation plot of the theoretical and experimental populations of different D0 arrangements derived from cryo-EM SPA analysis.

Authors propose that N-glycosylation plays a role in stabilizing the D0-up conformation. While this is interesting and could be tested by introducing a mutation into the S spike and performing a SPA, such an experiment was not done.

As per the suggestion of the Reviewer, we produced the recombinant PEDV PT52 S in HEK293 Freestyle cells for cryo-EM SPA in comparison with the PEDV PT52 S variant that harbors a T326I mutation, which nullifies the sequon for the N-glycosylation on Asn324. The rationale of the T326I design is explained above (response to Reviewer 1's first comment), and the results confirmed our hypothesis that the N-glycan on Asn324 helps promote the D0 in an up conformation. These new findings were described on pages 9-10 accompanied by the new Figure 5 and Figures S14.

The structure of S with neutralizing antibodies are not shown and it is not clear how this data will help in designing a more effective vaccine against PEDV.

With regard to the suggestion to study the structure of PEDV PT52 S in complex with a neutralizing antibody, there is currently no monoclonal antibody that forms a stable

complex with PEDV PT52 S, even for the E10E antibody as we mentioned in Discussion (page 13, line 15). Although Andrew Ward and colleagues reported the PEDV S structure in complex with antibody Fabs by negative-stain electron microscopy (<https://doi.org/10.1016/j.str.2020.12.003>), these antibody Fabs were derived from polyclonal antibodies whose identities were not well-established. Identification and developments of effective neutralizing antibodies against PEDV are indeed our long-term goals. We aim to identify neutralizing antibodies that bind to PEDV PT52 S with high specificity and affinity to enable detailed structural studies as suggested by the Reviewer.

In addition, no information is provided if any biosafety measures were observed during the propagation of the virus which is highly pathogenic in pigs.

We thank the Reviewer for the reminder to include the biosafety measure that has been put in place during our study. First of all, virus propagation and inactivation were carried out in a BSL-2 laboratory. Efficient virus inactivation was confirmed by measuring the cytopathic effect (CPE). To prepare the viral stocks, the PEDV PT52 strain was serially diluted in medium with a titer of 10^6 TCID₅₀/mL, and subsequently applied onto the Vero cells with a 90% confluency. The CPE of untreated PEDV on the Vero cell was determined in biological triplicates as a positive control, and the CPE of PBS treatment was determined as a negative control. The standard CPE of cells infected with PEDV is the virus-induced cellular morphological change, namely the formation of syncytial cells. Once the Vero cells showed visible cytopathic effects under light microscopy, it is considered to be infected by PEDV (see figure below).

Virus-induced Cytopathic effects of Vero cell. (A) Vero cell infected by 2 % Formaldehyde treated PEDV. No syncytial cell was observed suggesting inactivated PEDV failed to induce morphology change. (B) Vero cell infected by untreated PEDV. Syncytial cell (black arrows) was observed suggesting PEDV induce morphology change. (C) Vero cell without infection. No syncytial cell was observed.

Our results showed that all serial titers of inactivated PEDV failed to induce morphology change but not in untreated PEDV, confirming the effective inactivation of the PEDV samples used for cryo-ET and cryo-EM studies. We have now included the description for the biosafety measure on page 15, line 10 as follows:

“Complete inactivation of the virus was confirmed by rechallenging Vero cells.”

Reviewer #3 (Remarks to the Author):

In the manuscript submitted by Huang et al. the authors describe the application of cryo-ET and cryo-EM to investigate the structure and structural dynamics of a porcine epidemic diarrhea virus (PEDV) spike glycoprotein from a strain which is particularly lethal in neonatal piglets from either naïve or vaccinated herds, Pintung 52. This group of coronaviruses contains an additional domain not seen in other coronaviruses, such as SARS-CoV-2, the D0 domain. The authors investigate the dynamics of this domain and demonstrate that, unlike previous studies which select for particular conformations, there are multiple conformations of D0 possible on a single spike glycoprotein. In addition, the authors propose a model for the transition of the D0 domain from 'up' to 'down'. Finally, the authors investigate the N-linked glycans found on PEDV, using a separate strain of PEDV as the one investigated did not give high yields of protein. This analysis revealed

a range of different glycan processing states, as can be seen on a range of coronavirus spike proteins. Overall the study is an interesting study investigating the structure of infectious virions, and the observations are of importance to vaccine design for PEDV, from the perspective of D0 enabling viral immune escape.

My expertise is in the analysis of the glycosylation of viral glycoproteins, and as such I will focus most of my comments on this aspect of the manuscript, although, to my non-expert reading, the EM analysis appears well considered and follows a logical progression. My main comment on the EM data is that the authors correctly emphasize the presence of post-fusion spikes on infectious virions, and hypothesize that "The formation of the postfusion state could be attributed to the addition of trypsin in the cellular culture medium during the viral particle production". This raised a question about cleaved versus uncleaved spike protein on the mature virion. In HIV-1 Env and SARS-CoV-2, furin cleavage is required to produce infectious virions. Certain cell lines can contain low endogeneous furin and as a result, uncleaved material can be present on the surface of the virion. In HIV-1 Env, this uncleaved material can have distinct conformations compared to properly cleaved and folded Env.

I was wondering if the authors had considered the potential for uncleaved misfolded S protein to be present in their datasets and if they controlled for this? One potential way would be to dissociate the non-covalent interactions between S1 and S2 and investigate the proportion of spike proteins that remains intact, as this would suggest a lack of cleavage in the peptide backbone. There is potential for uncleaved material to be impacting the observed abundances of D0. If the authors could acknowledge this potential and, if feasible, perform a few additional experiments then that would be really interesting to know.

To address the Reviewer's question, we did not investigate whether the intact viral particles contained uncleaved misfolded S proteins biochemically, i.e., by SDS-PAGE and Western blotting. The cryo-ET data indicated the presence of postfusion S structures (Figure 2A), which corresponded to cleaved S proteins. As we did not have a structural template for a misfolded S protein for automated template matching, it would not be

possible to identify and quantify the misfolded S protein by cryo-ET. In the cryo-EM SPR analysis of intact viral particles, particle images that did not show well-defined trimer structures will be excluded during the workflow, so misfolded S protein could not be analyzed by such a procedure. For the recombinant S proteins purified from the IPEC-J2 or HEK293 Freestyle cells, they were purified by SEC to obtain well-folded trimers. The resulting samples showed a single band in the denatured SDS-PAGE, indicating that the S proteins were intact, not cleaved. Such a procedure will also exclude misfolded S protein from entering the cryo-EM analysis pipeline. In short, our structural analysis workflows were unlikely to identify cleaved S proteins and quantify their contributions to the different D0 arrangements.

From a glycobiology perspective, this reviewer certainly acknowledges the difficulties in obtaining sufficient material, and the use of a distinct spike glycoprotein is understandable. However, I feel that there are certain aspects to the glycan analysis that could be improved to provide a deeper insight into the glycosylation of PEDV. The authors describe the predominant glycan structure detected at each site in Figure 4. The materials and methods describe the process of quantification as "Identified glycopeptides were quantified based on the area under the curve of their extracted ion chromatogram (XIC-AUC)". As such there is likely more information that could be presented regarding the distribution of N-linked glycans on the S protein. Other coronavirus glycoprotein analyses demonstrate the different N-linked glycan compositions using XIC AUC analysis (<https://doi.org/10.1016/j.eng.2020.07.014> , <https://doi.org/10.1016/j.chom.2020.08.004>, doi: 10.1126/science.abb9983). This study would benefit from including the distributions of different N-linked glycans as N-linked glycan processing is an extremely heterogeneous process, and presenting more of the glycan data would provide more information about the structure-function relationship between the D0 and the glycans.

As per the suggestion by the Reviewer, we quantitatively analyzed the distributions of individual N-glycosylation sites of different recombinant S proteins in the revised manuscript. The analytical workflow was based on our recent work on SARS-CoV-2 S variants (<https://doi.org/10.1093/glycob/cwab102>). The results were described on page 9 with the accompanied revised Figure 4I, Figure S7, and Table S1. Figure 4I and Figure

S7 compared the distributions of different glycoforms of individual sites derived from different samples, and they showed the general consistency of the glycosylation patterns of the S proteins derived from different expression systems, namely the porcine IPEC-J2 cells and human HEK293 Freestyle cells, with the exception that the IPEC-J2 cell-derived S protein exhibited Gal α 1-3Gal that are absent in human as we commented on page 9, first paragraph as follows:

*“Additionally, several N-glycans, namely N62, N118, N216, N300, N344, N351, N556 and N667, contained terminal Hex-HexNAc units capped by an extra Hex residue, corresponding to the Gal α 1-3Gal epitope, which is absent in human (Error! Reference source not found.I and **Table S1**)^{53,54}.”*

Additionally, the authors should acknowledge the differences in glycosylation resulting from different cell lines. Many conditions contribute to the diverse range of N-linked glycans on mature human glycoproteins, and the production of virus/recombinant protein either in Vero cells (monkey origin) or HEK cells (human origin) will be different to that observed in pigs. For example, in addition to NeuAc, other mammals modify their N-linked glycans with NeuGc, which is not present to the same extent in HEK or Vero cells. Additionally Vero and HEK cells process glycans differently to one another (<https://doi.org/10.1021/acs.biochem.1c00279>) and it has been shown that Vero cell-derived glycans display relatively low levels of sialylation compared to other cells (DOI: 10.4149/av_2011_02_147). The authors should acknowledge these differences in the manuscript. If feasible the ideal outcome would be the production of virus in a porcine cell line and subsequent glycan analysis, although this may likely be beyond the scope of the project.

The Reviewer’s comment on the host cell-dependent glycosylation patterns is well-taken, and our breakthrough in generating the sequence-matched PEDV PT52 S sample in a recombinant form expressed in the native host cell, *i.e.*, the IPEC-J2 cell line, provided a biologically relevant source for detailed MS analyses. As the Reviewer rightly pointed out, the IPEC-J2 cells generated the Gal α 1-3Gal linkages in the recombinant S protein that were absent in HEK293F cell-derived S protein (Figure 4I, Figure S7, and Table S1).

Nevertheless, the differences in the terminal glycan structures did not impact the overall domain structures of the PEDV PT52 S protein (Figure 5 and Figure S14). Therefore, we could conclude that the difference in the sialylation in different expression systems made negligible contributions to the domain orientations of the S proteins. Nonetheless, these differences in glycosylation could potentially contribute host recognition and antigenicity. To clarify this point, we included a statement on page 13, lines 6-10 in the Discussion section to highlight our new findings that reads.

"Note that the recombinant PEDV PT52 S expressed in the HEK293F cells showed slightly different glycosylation pattern to that observed in the recombinant PEDV PT52 S expressed in the IPEC-J2 cells, particularly in the extent of terminal sialylation, since the IPEC-J2 cells would generate competing terminal Gala1-3Gal units that are absent in the HEK293F cell-derived recombinant protein (Figure 4I, Figure S7, and Table S1)"

We hope the Reviewer will find the addition satisfying.

Another point regarding the glycan analysis is the choice of sample to analyse. Whilst the selected strain has a high sequence homology to the strain studied by EM, there is an extra glycan site at position N723. As the authors state in the discussion: "In contrast, the D0-down conformation exhibited a cluster of N-glycans on Asn726, Asn873, and Asn118, which made extensive contacts with the D0 and S2 subunit". The presence of an N-linked glycan site so close to another will likely have impacts on the processing of the other, as can be seen on HIV-1 Env (<https://doi.org/10.1016/j.isci.2020.101711>). As a glycan site proximal to the D0 domain, I would encourage the authors to delete the N723 glycan site from the strain analysed, and investigate how the glycan processing compares. I feel that this would be a more appropriate model to use for the structures presented in this paper.

My final point is regarding the difficulty expressing and purifying coronavirus S protein from the strain analysed in the manuscript. Recent work building on previous coronavirus studies from the McLellan lab, including a 3.1 angstrom structure of PEDV, have produced high yields of spike protein using proline substitutions. Whilst not

applicable for the structural studies reported in this manuscript, this approach may enable protein to be produced corresponding to the Pintung 52 strain.

Major comments:

*To summarise the above, I feel that the current manuscript is a good fit for a specialist structural biology journal, however much more information can be gained from a deeper understanding of the glycan shield and how it influences the D0 conformation. **An ideal approach would be the production of a sequence matched protein in a porcine cell line.** If this is not possible then employing the mutations outlined above would be a good second option. If not then I think it is important to investigate the impacts of the deletion of the N723 site on the glycosylation of the remaining sites.*

We acknowledged the importance of having the sequence-matched recombinant sample derived from an appropriate host cell line for detailed glycosylation analyses. We initially attempted to obtain intact viruses from infected piglets for *in situ* structural studies and MS analysis of their glycoforms. We infected six piglets and sacrificed them 5 days post-infection. Although the piglets showed severe diarrhea due to PEDV infection, we could only obtain a small amount of virus sample that showed a positive signal of the most abundant nucleocapsid (N) protein in Western blotting. The quantity was insufficient for quantitative MS and cryo-EM studies (see figure below).

Characterization of PEDV Virion from Feces by Western Blotting. *The concentrated virion from both Piglets and Vero cells were analyzed by Western blot in lane 1 and 2 respectively. The bands corresponding to Spike, Nucleocapsid, and Membrane proteins are indicated with blue, yellow, and green triangles respectively.*

Nevertheless, we were delighted to have generated a stable cell line based on the IPEC-J2 cells for the recombinant PEDV PT52 S protein expression. The sample was compared with those derived from the HEK293 Freestyle cell line, which is more amenable to mutagenesis studies suggested by the Reviewers that were updated at the end of the Results section (page 10). As the reviewer suggested, we designed a synthetic gene to obtain a sequence-matched PEDV PT52 S protein construct to which a double proline mutation ($^{1076}\text{IL}^{1077} \rightarrow ^{1076}\text{PP}^{1077}$) was introduced in accordance with the design principle described by the McLellan lab to stabilize the prefusion state. The synthetic gene was subcloned to the mammalian expression vector pcDNA3.4-TOPO (Invitrogen), containing a trimerization domain (foldon) of T4 phage fibritin followed by c-Myc and a His6-tag at the C-terminus as described previously in our SARS-CoV-2 S protein studies. The recombinant PEDV PT52 S production in the IPEC-J2 and HEK293 Freestyle cell lines was extensively optimized to yield sufficient quantities for cryo-EM and MS analyses. Since the cryo-EM structures of PEDV PT52 S derived from the two cell lines were

identical (Figure S9), and the HEK293 Freestyle cell line is much more efficient for recombinant protein productions, we chose to use the HEK293 Freestyle cell line to carry the subsequent mutagenesis studies.

As described in the manuscript, we hypothesized that the N-glycan on Asn324 would stabilize D0-up conformation while three N-glycans on Asn118, Asn726, and Asn873 would stabilize the D0-down conformation. To test our hypothesis, we generated two constructs by introducing a single mutation, T326I, and triple N-to-Q mutations, i.e., N118Q, N726Q, and N873Q, to remove the respective N-glycans. The rationale for the N326I mutation instead of the direct N-to-Q mutation on Asn324 has been explained in the response to the Reviewer 1's first comment. We also confirmed the loss of the N-glycan in the cryo-EM maps of the N326I variant (Figures S14C). As described above, a comparison of the relative D0-up populations in the WT and the N326I variant confirmed our hypothesis that the N-glycan on Asn324 favors the D0-up conformation (Figure 5 and Figures S14).

In contrast to the success of the N326I variant, the triple mutant (N118Q, N726Q, and N873Q) did not show satisfactory expression levels of the recombinant S protein despite our best efforts (data not shown). As a result, we were unable to address the Reviewer's suggestion to examine the functional roles of these three N-glycans. Our speculation was that the three N-glycan-cluster plays an important role in the *de novo* folding of PEDV PT52 S during the quality control within ER. The loss of these N-glycans highly destabilized PEDV PT52 S leading to the greatly reduced expression level that prevented us from further characterizations. We hope the Reviewer will appreciate our efforts and find our new findings on the N326I variant acceptable for validating our hypothesis.

Despite the authors instructions, I was unable to access the raw data from the MASSive server using the reviewer instructions, this could be an error on my end, but I would encourage the authors to make the dataset public as soon as possible. The methods used for glycan analysis appear sound as written.

We thank the Reviewer's reminder. To access the server, please use the FTP server through the following link: <ftp://MSV000088544@massive.ucsd.edu> and the credentials below

User name: MSV000088544

Password: HC070225

Minor comments:

The introduction may benefit from some background regarding glycosylation and how it is important for protein structure.

We thank the Reviewer for this suggestion. We have now included two sentences at the end of the Introduction (page 4, line 20-24) that read

"Protein glycosylation plays a pivotal role in protein folding, pathogen-host recognition and immunity evasion⁴⁰. The integration of cryo-EM and MS to delineate the glycosylation patterns on glycoproteins, particularly for coronavirus S proteins, has emerged to become a powerful tool to help understand how individual glycans contribute to the biological functions^{30,41-44}"

We hope the Reviewer will find the addition satisfying.

In addition, some studies have reported potential receptors for PEDV, see ref 13 in the manuscript, "PEDV has been reported to utilize the aminopeptidase N protein (APN, also known as CD13) as a functional receptor"

Whether pAPN acts as a functional receptor for PEDV is controversial. A previous study uses an anti-pAPN antibody to block the interaction between pAPN and PEDV to imply that pAPN acts as a PEDV receptor (<https://doi.org/10.4142/jvs.2003.4.3.269>). One study generated a pAPN-expressing MDCK cell line, which showed enhanced PEDV infectivity as evidence that pAPN might serve as a PEDV receptor; another similar study used Vero cell as a model system to demonstrate the involvement of pAPN in mediating PEDV infection (<https://doi.org/10.1016/j.virol.2007.03.031>, PMID: 19634766,

<https://doi.org/10.4142/jvs.2003.4.3.269>). Nevertheless, more recent studies reported active infection of PEDV in a pAPN-knockout porcine (ST) cell line and two human cell lines (Huh7 and HeLa) as well as in pAPN-knockout pigs (<https://doi.org/10.1007/s12250-019-00127-y>, <https://doi.org/10.1038/s41598-019-49838-y>). These recent findings strongly suggest that pAPN is not a *bone fide* cellular receptor of PEDV but it may promote PEDV infectivity via its aminopeptidase activity (<https://doi.org/10.1099/jgv.0.000563>). The actual role of pAPN on the PEDV infection still needs to be further investigated. We have included the statements above in the revised Introduction, see page # and line #.

“Despite earlier reports that suggest porcine aminopeptidase N protein (pAPN, also known as CD13) to be a PEDV receptor²¹⁻²³, more recent findings observed PEDV infections in pAPN-knockout cell lines and pAPN-knockout pigs, suggesting that pAPN is not a bone fide PEDV receptor^{20,24,25}.”

We hope the Reviewer will find the addition satisfying.

Reviewers' Comments:

Reviewer #2:

Remarks to the Author:

The authors have addressed all my concerns. The additional experiments on PEDV PT52 S with the T326I mutation provide important mechanistic insights into the role of glycosylation and the movement of the D0 domain. This study brings important information on the structure of the alphacoronavirus PEDV S spike which is important for the development of effective neutralizing antibodies against PEDV.

Reviewer #3:

Remarks to the Author:

I appreciate the additional experiments performed by the authors and all of my comments have been appropriately addressed